# Video Diffusion Models Excel at Tracking Similar-Looking Objects Without Supervision

**Chenshuang Zhang**[1]    **Kang Zhang**[1]    **Joon Son Chung**[1]
**In So Kweon**[1]    **Junmo Kim**[1*]    **Chengzhi Mao**[2*]
KAIST[1], Rutgers University[2]

## Abstract

Distinguishing visually similar objects by their motion remains a critical challenge in computer vision. Although supervised trackers show promise, contemporary self-supervised trackers struggle when visual cues become ambiguous, limiting their scalability and generalization without extensive labeled data. We find that pre-trained video diffusion models inherently learn motion representations suitable for tracking without task-specific training. This ability arises because their denoising process isolates motion in early, high-noise stages, distinct from later appearance refinement. Capitalizing on this discovery, our self-supervised tracker significantly improves performance in distinguishing visually similar objects, an underexplored failure point for existing methods. Our method achieves up to a 6-point improvement over recent self-supervised approaches on established benchmarks and our newly introduced tests focused on tracking visually similar items. Visualizations confirm that these diffusion-derived motion representations enable robust tracking of even identical objects across challenging viewpoint changes and deformations. Project page: `https://chenshuang-zhang.github.io/projects/ted`.

## 1   Introduction

Imagine tracking one of two similar-looking deer walking in the forest (Figure 1(a)). Humans effortlessly resolve such visual ambiguities by relying on the distinct motion signatures of objects. This ability to perceive coherent objects through their unique temporal dynamics, even when static appearances are confounding, is fundamental. However, imbuing visual representations with this innate understanding of temporal dynamics, especially for tracking similar-looking objects, remains a significant challenge in computer vision [31, 23, 10, 51].

Many self-supervised methods [7, 23, 45], while good at learning intra-frame appearance features, fail when confronted with visually similar targets (see DIFT [45] in Figure 1(c)). Their Achilles' heel is the neglect of inter-frame temporal relationships. Even approaches that incorporate temporal signals through training objectives like cycle-consistency [32, 52, 24] often process frames independently at inference using 2D image encoders. This inherently limits their ability to model the continuous motion crucial for disambiguating similar objects in dynamic scenes (see CRW [24] and Spa-then-Temp [31] in Figure 1(c)).

In this paper, we show that representations for similar-looking object tracking do not need to be learned from scratch with intricate tracking-specific objectives. Instead, they can be repurposed within the internal workings of pre-trained video diffusion models [58, 2]. Unlike methods that view video as a sequence of isolated images [20, 23, 36], video diffusion models, by their very nature of generating coherent and realistic video, must implicitly capture the complex interplay of inter-frame dynamics. We find that the denoising process, particularly as it reconstructs motion from highly noisy states, already encodes a rich, motion-aware representation through its feature activations—a representation ripe for tracking without any explicit tracking supervision.

---

*Corresponding author. Junmo Kim <junmo.kim@kaist.ac.kr>, Chengzhi Mao <cm1838@cs.rutgers.edu>.

39th Conference on Neural Information Processing Systems (NeurIPS 2025).

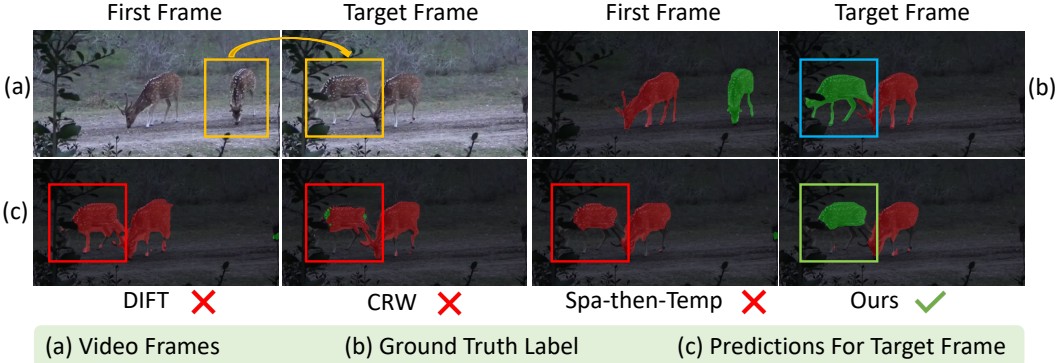

| First Frame | Target Frame | First Frame | Target Frame |
|---|---|---|---|

(a)

(c)

DIFF ✗   CRW ✗   Spa-then-Temp ✗   Ours ✓

(a) Video Frames   (b) Ground Truth Label   (c) Predictions For Target Frame

Figure 1: **Video label propagation on similar-looking objects.** State-of-the-art self-supervised trackers, such as DIFT [45], CRW [24] and Spa-then-Temp [31], often struggle when multiple objects look similar in a video. This failure is due to their exclusive reliance on appearance features. By dissecting and repurposing pretrained video diffusion models, we construct a feature that captures intra-frame motions in videos, allowing us to correctly track similar-looking objects, such as the deer highlighted by the green box in (c). In this figure, the green and red masks represent segmentation maps of different objects, while the blue, green, and red boxes highlight the ground truth regions, correctly predicted regions, and incorrectly predicted regions, respectively.

We introduce the **T**emporal **E**nhanced **D**iffusion tracking framework (TED), a simple yet remarkably effective approach that harnesses these latent diffusion features. TED synergizes the motion intelligence distilled by video diffusion models with conventional appearance features, enabling it to conquer the limitations of prior art [6, 23, 45] and robustly track visually indistinct objects (Figure 1(c)).

Experimental results show that our TED method outperforms 17 popular self-supervised models, achieving state-of-the-art performance in pixel-level object tracking. On the widely-used DAVIS-2017 benchmark [35], our TED significantly outperforms recent self-supervised methods [23, 45, 36, 31] by up to 6%. When evaluated on videos that include multiple similar-looking objects, our TED method achieves even larger improvement by up to 10%. Visualizations confirm that our representations encode differently for similar looking objects with different motion. Our approach also achieves significant improvement in other challenging scenarios, such as appearance-identical objects, real-world viewpoint changes, and object deformations.

## 2   Related Work

Learning video representations for temporal correspondence is crucial for visual tracking [46, 55, 31]. Due to limited annotations, recent studies have proposed various pretext tasks to learn representations in a self-supervised manner. We discuss related work below.

**Self-supervised representation learning from images.** Prior studies learn appearance features in video representations by training models on independent images [20, 6, 23, 45]. Some methods adopt instance discrimination as a pretext task, such as MoCo [20] and SimCLR [6]. SFC [23] improves further by integrating image-level and pixel-level cues for representation learning. DIFT [45] leverages knowledge from image diffusion models [40]. However, these methods only learn intra-frame appearance features, which fail in tracking visually similar objects (Figure 1(c)).

**Self-supervised representation learning from videos.** Some methods introduce temporal signals to model training, using two pretext tasks: cycle-consistency over time and frame reconstruction. Cycle-consistency task tracks a patch backward and forward in time to align its start and end points [32, 52, 24], while frame reconstruction aims to reconstruct pixels from adjacent frames [47, 28, 27]. Recent studies integrate temporal and spatial cues for training, such as Spa-then-Temp [31] and SMTC [36]. However, during inference, these models process frames independently using 2D image encoders, neglecting temporal context. Therefore, they fail to track similar-looking objects as in Figure 1(c).

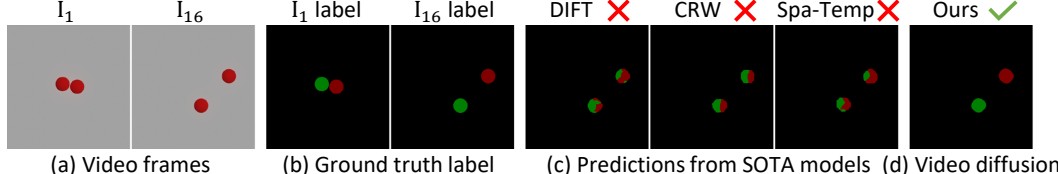

| $I_1$ | $I_{16}$ | $I_1$ label | $I_{16}$ label | DIFT ✗ | CRW ✗ | Spa-Temp ✗ | Ours ✓ |

(a) Video frames     (b) Ground truth label     (c) Predictions from SOTA models     (d) Video diffusion

Figure 2: **Our approach successfully tracks objects with identical appearances.** We conduct a controlled study, that we perform object label propagation on videos featuring two identical-looking and independently moving balls, with frames and their ground truth labels shown in (a) and (b). State-of-the-art methods [24, 31, 45] fail to distinguish these two balls, leading to incorrect predictions (c). In contrast, our approach accurately track both balls despite their identical appearance (d).

**Video object segmentation.** Supervised methods for video object segmentation [9, 57, 8] achieve impressive results but rely on large-scale annotated datasets for model training. For example, SAM2 [38] is trained on 50.9K videos with 35.5M masks. In contrast, our work addresses *self-supervised* tracking: no segmentation labels are used. Furthermore, models like SAM2 [38] use discriminative training objectives, whereas our work explores the inherent tracking capability of generative models. We find that video diffusion models can effectively track visually similar objects without any tracking-specific supervision, pointing to a promising direction for future trackers.

**Video diffusion models.** Diffusion models [21] have achieved great success in image generation [37, 43, 34, 42], such as Stable Diffusion [40] and ADM [14]. Video diffusion models further include temporal blocks for frame consistency [3, 48], with pioneering work Sora [4], I2VGen-XL [58], and Stable Video Diffusion [2]. Diffusion models have also been used in tasks like image classification [29, 12], semantic segmentation [1, 60, 53] and pose estimation [22, 16]. In contrast to these studies, we are the first to show that video diffusion models excel at tracking similar-looking objects without any tracking-specific training. Our finding that video diffusion models learn motions at high-noise stages also advances the understanding of video diffusion models.

**Track by diffusion models.** There has been recent interest in applying diffusion models to tracking [25, 59, 50]. Track4Gen [25] tackles point tracking by training video diffusion models on labeled point trajectories, whereas our work explores pretrained video diffusion models for object segmentation without any tracking-specific training. Diff-Tracker [59] is built on image diffusion models with additional motion encoders to learn temporal cues. By contrast, our work directly explores the built-in motion of pretrained video diffusion models without extra modules. VidSeg [50] performs instance-agnostic video semantic segmentation that cannot distinguish different objects in the same category. It also requires maintaining and updating an additional KNN classifier to learn temporal changes during tracking. By contrast, our approach can distinguish even similar-looking objects without extra components.

## 3   Challenges for Tracking Visually Similar Objects

**Task definition.** We focus on video label propagation task, which aims to transfer ground truth labels of the first frame (e.g., segmentation map) to subsequent frames [47]. The key is training models to obtain frame representation $\mathbf{R}$, which learns pixel-level correspondence among frames [23, 36, 31]. Due to limited annotations, prior studies train models in a self-supervised manner [24, 31], with pretext tasks like instance discrimination [20, 7, 23]. During inference, for each pixel in the current frame, the label is predicted by aggregating labels of its most similar pixels from previous frames, where the pixel similarity is computed using representation $\mathbf{R}$ (see Section 4.4 for details).

**Challenges for tracking similar-looking objects.** Label propagation for visually similar objects demands capturing robust motion signals, as appearance cues can become ambiguous and misleading. While prior studies excel at tracking objects using appearance features [23, 36, 31], they often fail when tracking multiple, similar-looking objects. We find this is due to their excessive dependence on appearance—a shortcut effective for distinct objects but a point of failure when objects are visually alike.

To illustrate this, we begin our investigation with a controlled toy experiment featuring two identical-looking, independently moving balls (Figure 2(a)). Given the ground truth segmentation map of each

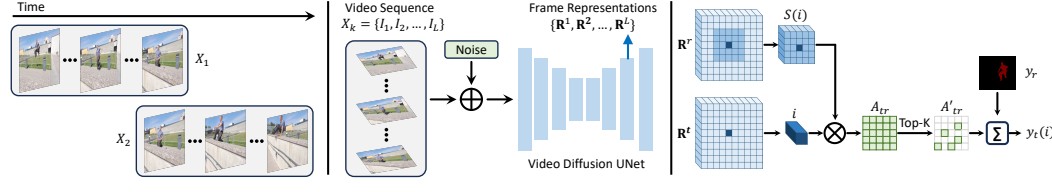

(a) Handle long videos with sliding windows    (b) Motion-aware representation for each window    (c) Track via label propagation

Figure 3: **Framework.** Our work tracks objects via video label propagation, which transfers ground truth label of the first frame to subsequent frames. As video diffusion models typically have a maximum input length, we first divide the long video into overlapping video windows (see (a)). For each window, we use video diffusion models to to extract frame representations that capture rich inter-frame motion features(see (b)). Specifically, our method uses the 3D UNet backbone that can process the entire video sequence along the temporal axis. Finally, to predict the label for a query pixel $i$ in the target frame ($\mathbf{R}^t$), we follow prior studies to aggregate the labels of its most similar pixels in previous frames (see (c); details in Section 4.4). We term our method **T**emporal **E**nhanced **D**iffusion tracking framework (**TED**). Experiments demonstrate that our TED improves tracking performance across diverse video scenarios, including those with similar-looking objects.

ball in the first frame (Figure 2(b), left), the task is to predict pixel-level labels of subsequent frames. Since the balls are identical, motion is the only signals that the trackers can rely on to make the right prediction. Figure 2(c) shows that state-of-the-art methods [45, 24, 31] struggle with object identity, leading to poor tracking. Moreover, we experiment on real-world videos with similar-looking objects. As shown in Figure 1(c), state-of-the-art trackers [45, 24, 31] fail to distinguish similar-looking deer when they swap positions. These findings highlight the difficulty of tracking multiple similar-looking objects in video label propagation.

## 4 Temporal-Enhanced Diffusion for Tracking

In Section 3, we show that state-of-the-art methods fail to track visually similar objects, highlighting the difficulty of self-supervised tracking when visual cues are ambiguous. To address this challenge, we propose a new Temporal-Enhanced Diffusion tracking framework (TED). We show that video representations for similar-objects tracking do not necessarily to be learned from scratch with tracking-specific objectives. Our TED leverages the motion intelligence from a pre-trained video diffusion model, enabling robust tracking of similar-looking objects.

In this section, we first introduce the tracking setup and video diffusion models. We then show how we obtain motion-aware representations without any tracking-specific objectives, and how to complement them with appearance features for further improvement. Finally, we show how these representations yield tracking results by label propagation.

### 4.1 Preliminaries: Tracking Setup and Video Diffusion Models

We focus on video label propagation task [47] as defined in Section 3. Following prior work [27], we aim to learn a frame representation, $\mathbf{R}^t$, for each frame $I_t$ of a video, such that the similarity between representations reflects the true correspondence of pixels across frames.

Our approach builds upon video diffusion models, which are often obtained by adding a temporal dimension to image diffusion models, using 3D architectures to capture temporal context. Video diffusion models are trained to generate realistic video sequences by learning to reverse a diffusion process [21, 39, 4]. This process involves adding Gaussian noise to a clean video $\mathbf{X}^0$ at different noise levels, indicated by step $\tau$. The model, $\epsilon_\theta$, is trained to predict the added noise at each step $\tau$, minimizing the following loss:

$$L = \mathbb{E}_{\mathbf{X}, \boldsymbol{\epsilon} \sim \mathcal{N}(0,1), \tau} \left[ \| \boldsymbol{\epsilon} - \epsilon_\theta(\mathbf{X}^\tau, \tau) \|_2^2 \right] \tag{1}$$

where

$$\mathbf{X}^\tau = \sqrt{\alpha_\tau} \mathbf{X}^0 + \sqrt{1 - \alpha_\tau}\, \varepsilon, \quad \varepsilon \sim \mathcal{N}(0,1) \tag{2}$$

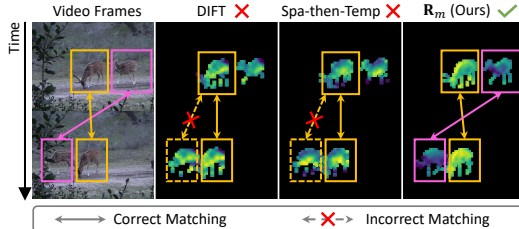

Figure 4: **Representation visualization.** We show a video with two similar looking deer swapping positions in the first column, and PCA results of model representations in column 2-4 (similar pixel colors indicate similar features). While state-of-the-art Spa-then-Temp [31] and DIFT [45] obtain similar features for both deer, our $\mathbf{R}_m$ distinguishes each deer by their different motions.

Here, $\alpha_\tau$ is a noise schedule parameter, with larger $\tau$ indicating higher noise levels. $\mathcal{N}(0, 1)$ denotes the Gaussian distribution. This training process forces the model to learn not only the visual content of individual frames, but also the coherent motion that connects them. Given a noisy video $\mathbf{X}^\tau$, the model obtains a cleaner $\mathbf{X}^{\tau-1}$ by removing the predicted noises in $\mathbf{X}^\tau$, termed denoising process. In this work, we study video diffusion models using the widely-used 3D UNet as $\epsilon_\theta$, which is built by inserting temporal layers, such as temporal attention and 3D convolution, into a 2D UNet [41]. Our work is also model-agnostic, which adapts to any pretrained video diffusion model that may not use a 3D UNet.

## 4.2 Space-time Learning for Motion-aware Representations

We begin by investigating the internal feature activations of video diffusion models. We find that high noise levels during the denoising process encode a rich representation of motion, while low noise levels primarily capture appearance information (see Figure 6 and Section 5.3). This discovery motivates us to a novel approach for object tracking. We leverage the internal feature activations ($\varepsilon_\theta$) at these high noise levels to extract robust motion cues.

Formally, given a video sequence $X = \{I_1, I_2, \ldots, I_N\}$, we first add noise to obtain $\mathbf{X}^\tau$ (following Equation 2). Then, we perform a single forward pass through the UNet of video diffusion models (UNet$_v$) to obtain features from $n_v$ layer, with entire video $\mathbf{X}^\tau$ as input (Figure 3(b)):

$$\mathbf{R}^1, \mathbf{R}^2, \ldots, \mathbf{R}^N = \text{UNet}_v(\mathbf{X}^\tau, n_v) \tag{3}$$

where $\tau$ represents the noise level. Crucially, unlike prior methods that compute $\mathbf{R}^t$ by independently processing each frame ($\mathbf{R}^t = F(I_t)$) using 2D image encoders, our video diffusion model's features $\mathbf{R}^1, \mathbf{R}^2, \ldots, \mathbf{R}^N$ incorporate temporal information through temporal attention and 3D convolutions. Consequently, each $\mathbf{R}^t$ represents not only the appearance of frame $I_t$, but also inter-frame motion dynamics captured within the representation from high-noise inputs, enabling effective tracking.

**Handling long videos: sliding window approach.** Video diffusion models typically have a maximum input length, $L$. To handle videos longer than that, inspired by temporal segment networks [49], we adopt a sliding window method, as shown in Figure 3(a). The video $X = \{I_1, I_2, \ldots, I_N\}$ is divided into multiple overlapping short video clips $\{X_k\}$ with window size $L$.

$$\begin{aligned}
X_1 &= \{I_1, I_2, \ldots, I_L\}, \\
X_2 &= \{I_{1+L-\text{overlap}}, I_{2+L-\text{overlap}}, \ldots, I_{L+L-\text{overlap}}\}, \\
&\vdots
\end{aligned} \tag{4}$$

where $0 \leq \text{overlap} < L$. By integrating the temporal knowledge from the 3D UNet, for each frame representation $\mathbf{R}^t$ in the clip, it encodes the temporal motions from all frames in the current video clip $X_k$. Overlapping frames further improve motion consistency among video clips.

**Visualization of motion-aware representations.** After obtaining representations from video diffusion models ($\mathbf{R}_m$), we study if $\mathbf{R}_m$ *can differentiate similar-looking objects.* We visualize both our representations $\mathbf{R}_m$ and representations from state-of-the-art methods [31, 45] in Figure 4, where two similar-looking deer swapping positions over time. We perform principal component analysis (PCA) [33] on two frames (denoted $s$ and $t$) for each model (e.g., $\tilde{\mathbf{R}}_m^s, \tilde{\mathbf{R}}_m^t = \text{PCA}(\mathbf{R}_m^s \parallel \mathbf{R}_m^t)$ for our $\mathbf{R}_m$). In Figure 4, similar pixel colors indicate similar representations. Figure 4 shows that prior methods [31, 45] capture similar features for different deer, indicated by similar colors. In contrast, our $\mathbf{R}_m$ learns clearly distinct features for each deer, shown as different color. These results highlight the superiority of our method in capturing object motions, enabling tracking similar-looking objects. Note that PCA is used only for visualization, and the original $\mathbf{R}_m$ is used for tracking.

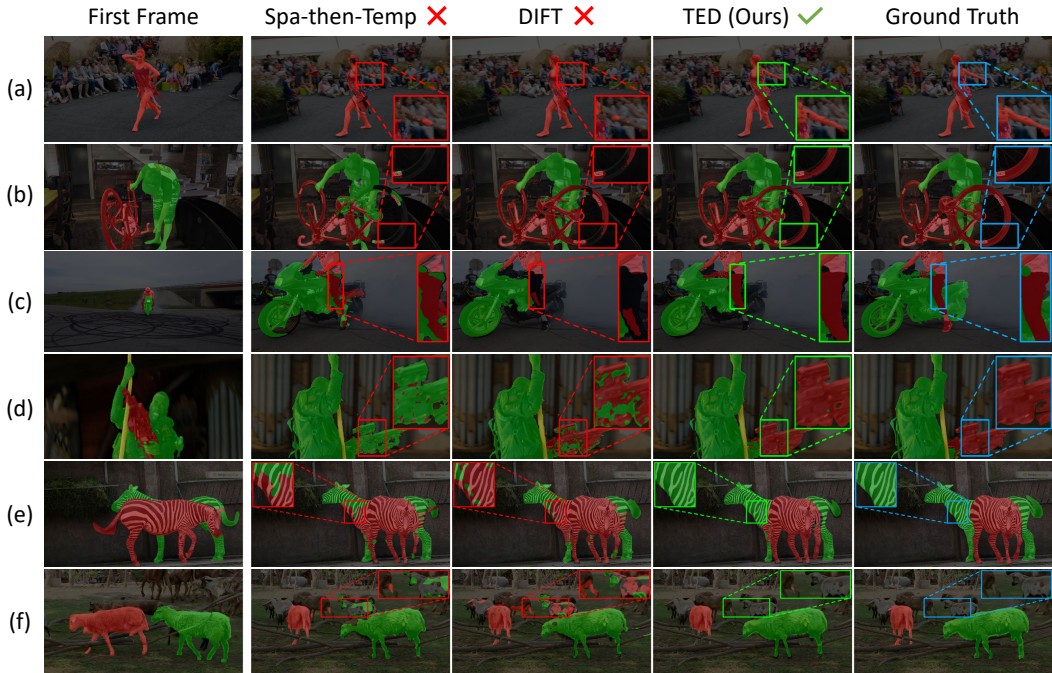

Figure 5: **Predictions for pixel-level object tracking.** We evaluate TED on the video label propagation task, comparing its predicted segmentation maps with those from state-of-the-art methods [31, 45]. Our TED consistently outperforms both methods [31, 45] on DAVIS (Figure a-d) and YouTube-Similar (Figure e-f) datasets, aligning with Table 1. Notably, our TED delivers more accurate predictions in scenarios with complex deformations (a) and viewpoint changes (b), while Spa-then-Temp [31] and DIFT [45] struggle with tracking completeness, e.g., the missing arm in (a). Our TED also achieves superior tracking in multi-object scenarios, such as interacting objects (c-d) and similar-looking objects (e-f). In contrast, Spa-then-Temp [31] and DIFT [45] have mislabeling issues, such as incorrect labels for the gun in (d) and misaligned labels for sheep in the background (f). These results show that our TED significantly improves tracking performance, highlighting the superiority of our motion-aware representations in tracking. (**Best viewed when zoomed in.**)

## 4.3 Motion Meets Appearance for Robust Tracking

While the motion-aware representations extracted from the video diffusion model ($\mathbf{R}_m$) are powerful, they are not the only source of useful information for tracking. Appearance cues remain important, particularly for distinguishing objects that are not identical. Inspired by the Two-Stream ConvNets [44], we combine the orthogonal motion ($\mathbf{R}_m$) features from video diffusion models and appearance ($\mathbf{R}_a$) features from a pre-trained image diffusion model [45]:

$$\mathbf{R}_f = \text{concat}\left(\lambda \cdot \frac{\mathbf{R}_m}{\|\mathbf{R}_m\|_2}, (1-\lambda) \cdot \frac{\mathbf{R}_a}{\|\mathbf{R}_a\|_2}\right) \tag{5}$$

where $\|\cdot\|$ denotes L2 normalization, and $\lambda$ is a weighting factor (between 0 and 1) that controls the relative importance of motion and appearance. Different from $\mathbf{R}_m$, which learns inter-frame features as defined in Equation 3, $\mathbf{R}_a$ is obtained by feeding each frame independently into the 2D UNet$_i$ of an image diffusion model. Specifically, for each frame $I_t$, we compute its appearance feature as $\mathbf{R}_a^t = \text{UNet}_i(I_t^\tau, n_i)$, where $I_t^\tau$ is computed by adding noise to $I_t$, and $n_i$ is the block index within the image diffusion model for feature representation. We refer to this combined approach using frame representation $\mathbf{R}_f$ for tracking as Temporal-Enhanced Diffusion tracking method (TED).

## 4.4 Tracking via Label Propagation

To perform tracking (i.e., label propagation), we follow the standard protocol used in previous work [52, 24, 23]. Given the ground truth labels in the first frame $I_1$, we use a recurrent method to propagate the labels to subsequent frames from $I_2$ to $I_N$, based on frame representations ($\mathbf{R}_f$).

Figure 3(c) shows how we predict the label for a query pixel $i$ in the target frame ($\mathbf{R}^t$). Define the first frame and the previous $m$ frames as reference frames, we first compute the pairwise similarities between pixel $i$ and pixels in the reference frames. In Figure 3(c), we show the case with one reference frame, with representation termed $\mathbf{R}^r$. Following prior studies [24, 23, 31], we restrict the similarity computation to a spatially local neighborhood $\mathcal{S}(i)$ around $i$. This yields a similarity matrix $A_{tr}$, where each element $A_{tr}(i, j)$ is the dot product of the representations of pixel $i$ in $\mathbf{R}^t$ and pixel $j$ in $\mathbf{R}^r$ (with $j \in \mathcal{S}(i)$). To identify the most similar pixels to $i$, we retain only the top-$K$ values from $A_{tr}$ to form $A'_{tr}$, setting all other values to zero. Finally, the label for pixel $i$ is predicted by aggregating the labels from its most similar pixels in the reference frames using a weighted sum:

$$y_t(i) = \sum_r \sum_{j \in \mathcal{S}(i)} A'_{tr}(i, j) \cdot y_r(j) \tag{6}$$

The pixel-level labels $y_t$ is computed in the representation space and then interpolated to match the size of video frames following [24, 23, 31]. We show the pseudocode of our TED in Appendix A.

## 4.5 Implementation Details

**Motion-aware representations $\mathbf{R}_m$.** Any pretrained video diffusion models can serve as our motion feature backbone. We default to the widely-used I2VGen-XL [58] and also explore Stable Video Diffusion [2]. Since I2VGen-XL supports up to 16 frames, longer videos are split into multiple 16-frame clips. We pass each clip through model's 3D UNet and extract features from the third block as $\mathbf{R}_m$. The diffusion step $\tau$ for computing model input $\mathbf{X}^\tau$ is chosen empirically.

**Appearance-aware representations $\mathbf{R}_a$.** Our TED framework supports any frame representations that learn appearance features. Different from $\mathbf{R}_m$ that takes the entire video sequence as model input (Section 4.2), we obtain $\mathbf{R}_a$ for each frame by inputting the image independently to the image encoder (i.e., $\mathbf{R}^t = F(I_t)$). We default to image-diffusion model ADM [14] as image encoder, extracting features from its eighth block as $\mathbf{R}_a$. We also test Stable Diffusion [39].

**Tracking via label propagation.** Our TED method uses $\mathbf{R}_f$, a fusion of $\mathbf{R}_m$ and $\mathbf{R}_a$, to obtain tracking results. We follow the setups of prior studies [24, 23, 45] for label propagation, with pseudocode and details in Appendix A and Appendix B.1.

# 5 Experiments

## 5.1 Experimental Setups

**Baselines.** Our TED advances self-supervised tracking without any labeled training data. We evaluate on video label propagation task, and compare against 17 state-of-the-art self-supervised methods.

`Self-supervised representation learning from images.` We evaluate 7 models that learn appearance features by training on independent images. We consider instance-discrimination methods, e.g., MoCo [20]. We also test SFC [23], a strong baseline that integrates image-level and pixel-level cues, and DIFT [45], which leverages knowledge from image diffusion models [40].

`Self-supervised representation learning from videos.` We benchmark 10 models that incorporate temporal cues to training through diverse pretext tasks. We evaluate on strong baselines trained for frame reconstruction (e.g., UVC [32]), cycle consistency (e.g., CRW [24]), and video contrastive learning (e.g., VFS [55]). We also include recent Spa-then-Temp [31] and SMTC [36].

**Datasets.** Our method uses pretrained diffusion models for tracking, without additional training. We benchmark on following test sets, with video examples in Appendix B.2. We follow prior studies [24, 36, 31] to report region similarity ($\mathcal{J}_m$) and contour accuracy ($\mathcal{F}_m$) for evaluation.

`Standard benchmark.` We follow previous work [32, 27, 52, 24, 36, 31] and evaluate on the widely-used DAVIS-2017 validation set [35], which contains 30 videos (2023 frames, 59 objects).

Table 1: **Results for pixel-level similar-looking object tracking task.** Our TED advances *self-supervised* tracking without any labeled data. We evaluate it on the video label propagation task against 17 state-of-the-art self-supervised methods. 'Temporal Train' indicates whether the method uses temporal signals during training. Colored numbers indicate the **best** results. Our TED achieves significant improvements across all datasets. On the widely-used DAVIS benchmark [35], our TED outperforms recent methods by up to 6%. On Youtube-Similar, featuring real-world similar-looking objects, our TED achieves an even larger gain of 10%. On Kubric-Similar, with two identical-looking, independently moving balls, TED reaches a high $\mathcal{J}_m$ of 87.2%, while most methods stay near 50%, equivalent to random guessing due to the objects' identical sizes. These results highlight the effectiveness of our TED in tracking similar-looking objects.

| Temporal Train | Method | DAVIS | | | Youtube-Similar | | | Kubric-Similar | | |
|---|---|---|---|---|---|---|---|---|---|---|
| | | $\mathcal{J}\&\mathcal{F}_m(\uparrow)$ | $\mathcal{J}_m(\uparrow)$ | $\mathcal{F}_m(\uparrow)$ | $\mathcal{J}\&\mathcal{F}_m(\uparrow)$ | $\mathcal{J}_m(\uparrow)$ | $\mathcal{F}_m(\uparrow)$ | $\mathcal{J}\&\mathcal{F}_m(\uparrow)$ | $\mathcal{J}_m(\uparrow)$ | $\mathcal{F}_m(\uparrow)$ |
| | InstDis [54] | 66.4 | 63.9 | 68.9 | - | - | - | - | - | - |
| | MoCo [20] | 65.9 | 63.4 | 68.4 | 48.0 | 48.5 | 47.4 | 56.6 | 51.6 | 61.6 |
| | SimCLR [6] | 66.9 | 64.4 | 69.4 | 37.5 | 36.9 | 38.1 | 55.6 | 50.3 | 60.9 |
| ✗ | BYOL [19] | 66.5 | 64.0 | 69.0 | 47.1 | 47.7 | 46.5 | 54.8 | 49.2 | 60.5 |
| | SimSiam [7] | 67.2 | 64.8 | 68.8 | 47.4 | 47.9 | 47.0 | 58.4 | 52.6 | 64.1 |
| | SFC [23] | 71.2 | 68.3 | 74.0 | 55.5 | 55.3 | 55.7 | 47.7 | 43.1 | 52.3 |
| | DIFT [45] | 75.7 | 72.7 | 78.6 | 60.7 | 59.8 | 61.7 | 55.1 | 52.7 | 57.6 |
| | Colorization [47] | 34.0 | 34.6 | 32.7 | - | - | - | - | - | - |
| | TimeCycle [52] | 48.7 | 46.4 | 50.0 | 39.8 | 41.3 | 38.2 | 50.6 | 44.0 | 57.2 |
| | CorrFlow [28] | 50.3 | 48.4 | 52.2 | 39.6 | 40.0 | 39.3 | 32.6 | 27.0 | 38.3 |
| | UVC [32] | 60.9 | 59.3 | 62.7 | 49.7 | 49.8 | 49.7 | 56.9 | 51.3 | 62.6 |
| | VINCE [17] | 65.2 | 62.5 | 67.8 | 44.9 | 45.4 | 44.3 | 54.1 | 48.5 | 59.7 |
| ✓ | MAST [27] | 65.5 | 63.3 | 67.6 | - | - | - | - | - | - |
| | CRW [24] | 67.6 | 64.8 | 70.2 | 52.0 | 52.3 | 51.6 | 54.9 | 49.7 | 60.1 |
| | VFS [55] | 68.9 | 66.5 | 71.3 | 57.3 | 57.1 | 57.5 | 44.2 | 38.5 | 49.9 |
| | SMTC [36] | 73.0 | 69.4 | 76.6 | 57.5 | 57.2 | 57.9 | 68.6 | 64.7 | 72.5 |
| | Spa-then-Temp [31] | 74.1 | 71.1 | 77.1 | 59.6 | 59.2 | 60.1 | 48.9 | 44.0 | 53.8 |
| ✓ | **TED (Ours)** | **77.6** | **74.4** | **80.8** | **66.0** | **65.1** | **67.0** | **90.2** | **87.2** | **93.1** |

`Real-world similar-looking benchmark.` We introduce Youtube-Similar, including 28 videos featuring similar-looking objects from Youtube-VOS [56], totally 839 frames and 69 objects.

`Controlled identical-object benchmark.` In real-world videos, visually similar objects can still differ due to factors like gestures. To eliminate these variations, we introduce Kubric-Similar, including 30 videos (480 frames, 60 objects) in which two identical-looking balls move independently. The dataset is generated by Kubric simulator [18], with random ball colors, sizes, and motions.

## 5.2 Experimental Results

**Quantitative results.** We compare our TED method with 17 self-supervised methods in Table 1. Our TED achieves the **state-of-the-art** tracking performance on **all** datasets. On the standard DAVIS dataset, our TED significantly outperforms recent methods by up to 6%, such as SFC [5] by 6.4%, SMTC [36] by 4.6%, Spa-then-Temp [31] by 3.5% and DIFT [45] by 1.9%. By introducing motion-aware features from video diffusion models, our TED achieves an even greater improvement when tracking similar-looking objects on Youtube-Similar, such as Spa-then-Temp [31] by 6.4% and DIFT [45] by 5.3%. On Kubric-Similar that includes identical objects, many methods achieve a $\mathcal{J}_m$ around 50%, no better than random guessing due to identical sizes of two balls. By contrast, our TED achieves a high $\mathcal{J}_m$ of 87.2%. These improvements highlight the effectiveness of our method in object tracking, even for challenging settings with multiple similar-looking objects.

**Visualizations.** Figure 5 compares our tracking results with state-of-the-art methods on the DAVIS dataset (a-d) and YouTube-Similar (e-f). Our TED approach significantly outperforms prior methods, aligning with Table 1. Our TED effectively handles complex deformations (Figure 5(a)) and viewpoint changes (Figure 5(b)), while Spa-then-Temp [31] and DIFT [45] struggle with elements like the human arm (Figure 5(a)). Our TED also excels in multi-object scenarios, such as interacting objects (Figure 5(c-d)) and similar-looking objects (Figure 5(e-f)). By contrast, Spa-then-Temp [31] and DIFT [45] often confuse different objects, leading to incorrect tracking results. For example, in Figure 5(d), Spa-then-Temp mislabels a gun as a human and DIFT shows significant contour errors. In Figure 5(f), both Spa-then-Temp and DIFT mistakenly assign the target label to a background

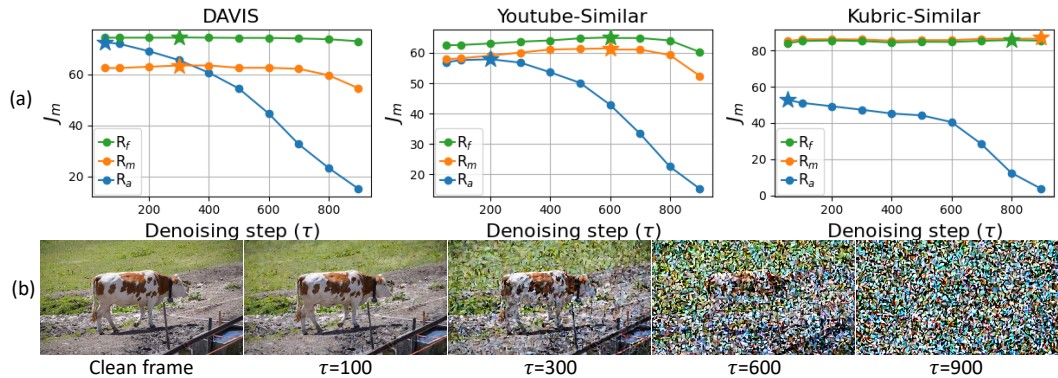

Figure 6: **Tracking results under different denoising steps.** We evaluate tracking performance using model inputs $\mathbf{X}^\tau$ at various denoising steps $\tau$, where larger $\tau$ indicates more noise (see (b)). The performance of appearance features $\mathbf{R}_a$ degrade significantly as $\tau$ increases, while our motion feature $\mathbf{R}_m$ maintains high tracking accuracy even with a large $\tau$. Notably, $\mathbf{R}_m$ peaks at $\tau$=600 on Youtube-Similar and $\tau$=900 on Kubric-Similar, where appearance cues are almost available. These results reveal that video diffusion models can learn object motions from highly noisy inputs, enabling effective, motion-aware tracking.

sheep. These results demonstrate that our TED significantly outperforms prior methods across various scenarios, highlighting the superiority of motion-aware features in our work.

## 5.3 Analysis and Ablation Studies

**The impact of diffusion features from different noisy levels on motion-based tracking.** We evaluate tracking performance using model inputs $\mathbf{X}^\tau$ at various denoising steps $\tau$ (see Equation 2 and Equation 3), as shown in Figure 6. We will show that video diffusion models capture object motions even at early denoising steps when the input $\mathbf{X}^\tau$ is highly noisy.

Figure 6(a) shows that the tracking performance of appearance feature $\mathbf{R}_a$ drops significantly with larger $\tau$ (e.g., $\tau \geq 600$). This is because $\mathbf{X}^\tau$ is heavily corrupted at high noise levels, as shown in Figure 6(b), thus appearance features are almost unavailable. In contrast, our motion feature $\mathbf{R}_m$ achieves high tracking results at high noise levels. Interestingly, $\mathbf{R}_m$ even achieves its best performance (marked by a star in the figure) at $\tau$=600 on Youtube-Similar and $\tau$=900 on Kubric-Similar, when $\mathbf{R}_a$ almost fails. While motion features are crucial for identifying similar-looking objects, appearance features provide fine-grained details for accurate segmentation. This explains why our fused representation $\mathbf{R}_f$ outperforms motion-only features $\mathbf{R}_m$ in real-world data, highlighting the benefit of jointly leveraging motion and appearance cues for tracking.

An interesting question is: how do video diffusion models learn object motions with highly noisy input $\mathbf{X}^\tau$? With loss defined in Equation 1, diffusion models are trained to reconstruct clean input from its noisy counterparts. To achieve this goal, they solve different tasks at different noise levels [11]. When $\mathbf{X}^\tau$ is highly corrupted at high noise levels, video diffusion models are trained to solve the hard task that learns coarse-grained signals in the video, such as motion (e.g., changes of object positions among frames). Therefore, its representation $\mathbf{R}_m$ encodes rich motion information that enables effective tracking of similar-looking objects. When input $\mathbf{X}^\tau$ is less noisy, diffusion model is trained to denoise appearance details, where motion features are also learned but may not be so prioritized, leading to performance decrease at low noise levels. Our analysis and results

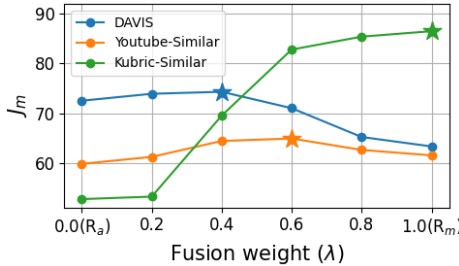

Figure 7: **Fusion weight** ($\lambda$). Our method integrates the advantages of motion and appearance features. For dataset where visual clues are ambiguous, such as Kubric, more video diffusion features are important in improving the accuracy.

provide new insights into both tracking and video diffusion models.

**The effect of coefficients for combining motion and appearance.** Figure 7 shows tracking accuracy with varying fusion weight $\lambda$ (see Equation 5), where $\lambda=1$ gives $\mathbf{R}_f = \mathbf{R}_m$ while $\lambda=0$ gives $\mathbf{R}_f = \mathbf{R}_a$. On Kubric-Similar with visually identical balls, motion features solely are sufficient for successful tracking. Our results align with this expectation by achieving the best result with $\lambda=1.0$. On real-world DAVIS and Youtube-Similar, our $\mathbf{R}_f$ performs best with a moderate $\lambda$ value around 0.5. These results show that our $\mathbf{R}_f$ effectively integrates the advantages of motion and appearance features in complex scenarios, outperforming the case of using either $\mathbf{R}_m$ or $\mathbf{R}_a$ alone.

**The effect of layers for feature representation.** We extract representations from internal layers of video and image diffusion models for tracking, with block indices denoted as $n_v$ and $n_i$ as in Section 4. Our framework is agnostic to specific layers. Motion and appearance features can be taken from different layers, and the optimal layers for the two backbones do not need to be the same. Following [45], we use the decoder representations from UNet. We report the tracking results using $\mathbf{R}_m$ alone from different decoder blocks in Table 2. Table 2 shows that the medium block (block 3) yields the best performance among all blocks on the DAVIS dataset.

Table 2: **Block indexes.** Motion representation $\mathbf{R}_m$ achieves the best tracking results when extracted from the third block of pretrained I2VGen-XL [58].

| Block | $\mathcal{J}\&\mathcal{F}_{\mathrm{m}}(\uparrow)$ | $\mathcal{J}_{\mathrm{m}}(\uparrow)$ | $\mathcal{F}_{\mathrm{m}}(\uparrow)$ |
|---|---|---|---|
| 1 | 24.8 | 28.2 | 21.4 |
| 2 | 47.6 | 52.7 | 42.5 |
| **3** | **66.3** | **63.4** | **69.1** |
| 4 | 31.5 | 27.2 | 35.8 |

**The impact of overlapping frames on tracking.** Motion features are crucial for successful tracking of similar-looking objects. Figure 8 shows the tracking accuracy on Youtube-Similar versus overlapping frames ($l$) among video clips. Compared to the non-overlapping case ($l=0$), introducing overlapping frames ($l>0$) achieves higher tracking accuracy due to improved motion consistency among video clips. At the same time, a small $l$ (e.g., $l=2$) is sufficient for good performance since tracking accuracy improves marginally with higher values of $l$. These results highlight the importance of accurate inter-frame motions in frame representations when tracking similar-looking objects.

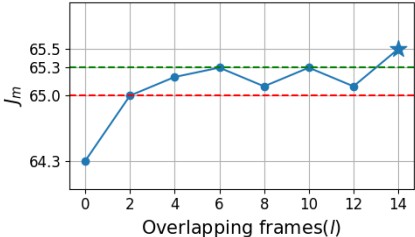

Figure 8: **Overlapping frames among video clips** ($l$). A small $l$ (e.g., $l=2$) is sufficient for boosting tracking accuracy, highlighting the importance of using multiple input frames in capturing motion clue for our method.

**The effect of diffusion models.** We evaluate our TED using representations from different diffusion models on DAVIS dataset, as shown in Table 3. Our TED achieves the best tracking results using motion features from I2VGen-XL [58] and appearance features from ADM [14], which are used by default in this work.

Table 3: **Pretrained diffusion models.**. Our TED achieves the best tracking results using representations from I2VGen-XL [58] and ADM [14].

| $\mathbf{R}_m$ | $\mathbf{R}_a$ | $\mathcal{J}\&\mathcal{F}_{\mathrm{m}}(\uparrow)$ | $\mathcal{J}_{\mathrm{m}}(\uparrow)$ | $\mathcal{F}_{\mathrm{m}}(\uparrow)$ |
|---|---|---|---|---|
| SVD [2] | SD [39] | 71.5 | 68.9 | 74.1 |
| SVD [2] | ADM [14] | 76.6 | 73.6 | 79.7 |
| I2VGen [58] | SD [39] | 71.7 | 69.0 | 74.5 |
| I2VGen [58] | ADM [14] | **77.6** | **74.4** | **80.8** |

**Computation cost analysis.** Our method has 20% more computation time than generative model method DIFT [45] on DAVIS videos and 89% than the popular self-supervised discriminative model method SFC [23]. For memory, our method uses 64% more than DIFT and 705% more than SFC. See more details in Appendix B.3).

## 6 Conclusion

We demonstrate that video diffusion models excel at tracking similar-looking objects without any task-specific training. We show that pre-trained video diffusion models possess an inherent, previously unrecognized ability to encode motion information at high noise levels. Rather than designing complex architectures or training objectives for tracking, we simply extract this readily available motion representation, achieving state-of-the-art tracking performance. Our approach achieves significant improvement over prior methods in diverse scenarios, such as challenging viewpoint changes and deformations. Our work opens new avenues for leveraging the latent capabilities of diffusion models beyond generation.

## Acknowledgments and Disclosure of Funding

This work was supported by Institute of Information & communications Technology Planning & Evaluation (IITP) grant funded by the Korea government(MSIT) (No.RS-2022-II220184, Development and Study of AI Technologies to Inexpensively Conform to Evolving Policy on Ethics), and by the ITRC (Information Technology Research Center) grant funded by the Korean government (IITP-2025-RS-2023-00259991).

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

## A  Pseudocode of Our Temporal Enhanced Diffusion Tracking Method (TED)

We provide the pseudocode of our Temporal Enhanced Diffusion tracking method (TED) in Algorithm 1. For clarity, we denote the process of obtaining noisy input $\mathbf{X}^\tau$ for video diffusion model in Equation 2 of Section 4.1 as the function AddNoiseVD. We also term the similar process for image diffusion models that adds noise to image inputs as AddNoiseID.

## B  Experimental Setups and Results

### B.1  Tracking Setups for Label Propagation

We follow the experimental setups of prior studies [23, 45, 31] for video label propagation, as summarized in Table 4.

Table 4: Experimental setups of TED for video label propagation.

| Dataset | Video diffusion | | | Image diffusion | | | Fusion weight | Softmax temp | Propagation radius | k for top-k |
|---|---|---|---|---|---|---|---|---|---|---|
| | Model | Timestep | Block | Model | Timestep | Block | | | | |
| DAVIS | I2VGen-XL | 300 | 3 | ADM | 51 | 8 | 0.4 | 0.2 | 15 | 10 |
| Youtube-Similar | I2VGen-XL | 600 | 3 | ADM | 51 | 8 | 0.6 | 0.1 | 15 | 10 |
| Kubric-Similar | I2VGen-XL | 900 | 3 | ADM | 51 | 8 | 1.0 | 0.1 | 15 | 10 |

### B.2  Datasets

Our method applies pretrained diffusion models for tracking, without additional training. We introduce the test sets in Section 5.1 and show video examples from each dataset in Figure 9.

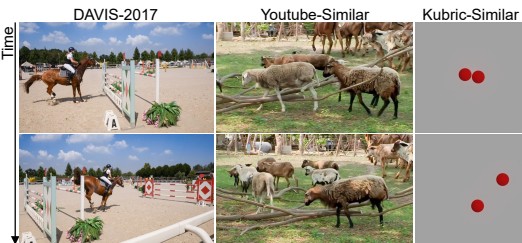

Figure 9: **Video examples from test sets.** Following prior studies [24, 36, 31, 45], we evaluate on the standard DAVIS-2017 benchmark [35] (first column). To evaluate tracking on visually similar objects, we introduce Youtube-Similar (second column), a real-world test set with similar-looking objects, and Kubric-Similar (third column), a controlled set with identical objects.

To ensure the quality of Youtube-Similar, three graduate students were hired to manually select videos from Youtube-VOS [56] according to the following rubrics. Only the videos that all annotators find qualified are included in the final YouTube-Similar dataset. The first rubric is object similarity. To create a dataset with similar-looking objects, we first select videos that contain at least two objects belonging to the same category from YouTube-VOS. The second rubric is video filtering. We exclude static videos from the pool obtained in the first stage. In such videos, spatial appearance features can serve as a shortcut for tracking, which may influence our evaluation of the motion features learned in model representations.

### B.3  Computational Cost Analysis

We compare computation cost with prior methods in Table 5, tested on a single A100 GPU using DAVIS videos. Our full model achieves the highest accuracy (77.6%) using a time of 682 ms per frame, compared to DIFT (75.7%, 566 ms) and SFC (71.2%, 360 ms).

For each video, a technique to boost accuracy in our full method is averaging representations computed from multiple noisy inputs. We also provide an efficient variant by removing this averaging step, which runs at 521 ms and achieves 77.2% accuracy, still significantly outperforming prior methods. Our efficient version achieves a tradeoff between tracking accuracy and efficiency.

Our work is the first to show that video diffusion models can track similar-looking objects without tracking-specific training. Our finding that motion features are learned at high-noise stages also provides new insight into video diffusion models. Additionally, our method is compatible with

---

**Algorithm 1:** **T**emporal **E**nhanced **D**iffusion Tracking (TED)

---

**Input:** Video frames $I_1, I_2, \ldots, I_N$; Ground-truth label $Y_1$ for $I_1$; Video diffusion model $\text{UNet}_v$; Image diffusion model $\text{UNet}_i$; Denoising steps: $\tau_v$ (video diffusion) and $\tau_i$ (image diffusion); Block index for features: $n_v$ (video diffusion) and $n_i$ (image diffusion); Fusion weight $\lambda$.

**Output:** Label predictions $Y_2, Y_3, \ldots, Y_N$ for frames $I_2, \ldots, I_N$.

1 Initialize a queue $Q \leftarrow \emptyset$ for storing representations and labels of reference frames;
2 Let $L$ be the maximum input length for $\text{UNet}_v$ and $l$ be the number of overlapping frames between video clips;
3 Divide the entire long video to $ClipNumber = \lfloor (N - L)/(L - l) \rfloor + 1$ video clips using sliding window approach.
4 **for** $k = 0$ **to** $ClipNumber - 1$ **do**
5   Define current video clip $X_k = \{I_{1+k(L-l)}, \ldots, I_{L+k(L-l)}\}$;

6   $\boxed{\textbf{Step 1: Compute Frame Representations}}$

7   (a) *Motion-aware* $\mathbf{R}_m$ : Compute $\mathbf{R}_m$ in one forward pass of $\text{UNet}_v$:
8     $\mathbf{R}_m^{1+k(L-l)}, \ldots, \mathbf{R}_m^{L+k(L-l)} = \text{UNet}_v(\text{AddNoiseVD}(X_k, \tau_v), n_v)$;
9   (b) *Appearance-aware* $\mathbf{R}_a$: Compute $\mathbf{R}_a$ in multiple forward pass of $\text{UNet}_i$:
10     For each frame $I_t \in X_k$, compute $\mathbf{R}_a^t = \text{UNet}_i(\text{AddNoiseID}(I_t, \tau_i), n_i)$;
11   (c) *Fused* $\mathbf{R}_f$: For each frame $I_t \in X_k$, compute fused representation:
   $\mathbf{R}_f^t = \text{concat}\left( \lambda \frac{\mathbf{R}_m^t}{\|\mathbf{R}_m^t\|_2}, (1-\lambda) \frac{\mathbf{R}_a^t}{\|\mathbf{R}_a^t\|_2} \right)$

12   $\boxed{\textbf{Step 2: Predict Tracking Labels}}$

13   **if** $k = 0$ **then**
14    Resize label $Y_1$ to match the size of $\mathbf{R}_f$, termed as as $y_1$. Add $(\mathbf{R}_f^1, y_1)$ to $Q$;

15   **for** *each frame $I_t \in X_k$* **do**
16    **if** *$(\mathbf{R}_f^t, y_t)$ is already in $Q$* **then**
17     continue;

18    **for** *each query pixel $i$ in $\mathbf{R}_f^t$* **do**
19     **for** *each pixel $j$ from each reference frame $\mathbf{R}_r \in Q$* **do**
20      **if** *$j$ locates in the spatial neighborhood of pixel $i$ ($\mathcal{S}(i)$)* **then**
21       Compute similarity score $A_{tr}(i,j) = \text{DotProduct}(\mathbf{R}_f^t(i), \mathbf{R}_f^r(j))$ ;

22     Identify the most similar pixels to $i$ by retaining the top-$K$ values in $A_{tr}$ and setting others as zero, obtaining $A'_{tr}$;
23     Predict the label of pixel $i$ by: $y_t(i) = \sum_{r \in Q} \sum_{j \in \mathcal{S}(i)} A'_{tr}(i,j)\, y_r(j)$
24    Add $(\mathbf{R}_f^t, y_t)$ to $Q$;
25    **if** *Size(Q) equals the maximum allowed reference frames* **then**
26     remove the oldest entry from $Q$;
27    Interpolate $y_t$ to the original frame size to obtain $Y_t$;

28 **return** $Y_2, Y_3, \ldots, Y_N$;

---

Table 5: **Computation cost analysis.** We report the tracking accuracy ($\mathcal{J}\&\mathcal{F}_\text{m}$) and computation cost per frame on DAVIS. Our full method achieves the highest accuracy, while our efficient version achieves a tradeoff between tracking accuracy and efficiency.

| Method | Accuracy | Time (ms) | Memory (GB) |
|---|---|---|---|
| SFC [23] | 71.2 | 360 | 1.9 |
| DIFT [45] | 75.7 | 566 | 9.3 |
| Ours (Efficient) | 77.2 | 521 | 11.8 |
| Ours (Full) | 77.6 | 682 | 15.3 |

acceleration techniques such as FlashAttention [13] and quantization [30] for further speedup. We leave further efficiency optimization for future work.

## B.4 Results on More Tasks and Datasets

We conduct experiments for human pose tracking on JHMDB dataset [26] and human part tracking on VIP dataset [61], as shown in Table 6. We also evaluate our method on YouTube-VOS [56] and MOSE [15] for video object segmentation, as shown in Table 7. Experimental results show that our method consistently improves tracking accuracy across diverse datasets and tracking tasks.

Table 6: Results on JHMDB for human pose tracking and VIP dataset for human part tracking.

| Dataset | JHMDB | | VIP |
| --- | --- | --- | --- |
| | PCK@0.1($\uparrow$) | PCK@0.2($\uparrow$) | mIoU ($\uparrow$) |
| SFC [23] | 61.9 | 83.0 | 38.4 |
| Spa-then-temp [31] | 66.4 | 84.4 | 41.0 |
| DIFT [45] | 63.4 | 84.3 | 43.7 |
| **TED (Ours)** | **68.3** | **85.8** | **44.2** |

Table 7: Results on YouTube-VOS and MOSE datasets for video object segmentation.

| | YouTube-VOS | | | MOSE | | |
| --- | --- | --- | --- | --- | --- | --- |
| | $\mathcal{J}\&\mathcal{F}_{\mathrm{m}}(\uparrow)$ | $\mathcal{J}_{\mathrm{m}}(\uparrow)$ | $\mathcal{F}_{\mathrm{m}}(\uparrow)$ | $\mathcal{J}\&\mathcal{F}_{\mathrm{m}}(\uparrow)$ | $\mathcal{J}_{\mathrm{m}}(\uparrow)$ | $\mathcal{F}_{\mathrm{m}}(\uparrow)$ |
| DIFT [45] | 70.5 | 68.2 | 72.7 | 34.5 | 28.9 | 40.1 |
| **TED (Ours)** | **71.1** | **68.9** | **73.4** | **35.6** | **30.1** | **41.1** |

## C  Discussions

**Limitations and future work.** Although our approach achieves significant tracking improvement across various scenarios. it comes with certain limitations. As discussed in Section 5.3 and Appendix B.3, using video and image diffusion models for tracking increases computational cost compared to previous methods. However, our work aim to show that video diffusion models can effectively track similar objects without tracking-specific training, suggesting a new direction for future trackers. Our finding that motions are learned at high-noise stages also advances the understanding of video diffusion models. One promising research direction is distilling motion intelligence from video diffusion models into smaller models for more efficient tracking. We leave further efficiency improvements for future work.

**Broader impacts.** Our work leverages the motion intelligence from video diffusion models and achieves the state-of-the-art tracking performance without task-specific supervision, reducing reliance on costly labeled data. This greatly benefit various applications, such as robotics and autonomous driving. Since our method leverages pretrained diffusion models for tracking, its performance may reflect biases present in those models, such as the underrepresentation of certain video types. We believe that mitigating bias in diffusion models is a promising research direction. We hope our work encourages further studies in this field, which benefit not only generative tasks but also perception tasks such as tracking.

