# OpenReview forum: "Video Diffusion Models Excel at Tracking Similar-Looking Objects Without Supervision"
_NeurIPS.cc/2025/Conference — NeurIPS 2025 poster_

### Official Review · Reviewer_aXR3 · 2025-06-23

**Clarity:** 3
**Significance:** 3
**Originality:** 4
**Rating:** 5
**Confidence:** 3

**Summary:**

This paper utilizes the pre-trained video diffusion model as a feature extractor for video frames, where each frame representation can be encoded with rich motion knowledge and appearance information with various noise levels. By combining the motion feature and appearance feature for each frame, the proposed method consistently improves label propagation for tracking on three benchmarks without supervised fine-tuning.

**Questions:**

1. Figure 1 is unclear. Please indicate the difference between the blue, green, and red boxes and the green and red masks in Figure 1.c in the caption.
2. Please also refer to the weakness.

**Ethical Concerns:**

["NO or VERY MINOR ethics concerns only"]

**Final Justification:**

This paper presents a novel and conceptually compelling use of video diffusion models for object tracking, achieving state-of-the-art performance without supervised training and demonstrating strong cross-backbone generalization. The rebuttal addressed my main concerns well. The authors clarified the role of motion versus appearance features: at higher noise levels, motion features dominate and are sufficient for successful tracking, while appearance features remain valuable for fine-grained segmentation. They also clarified Figure 1(c) and will revise the caption accordingly. Although computational cost is non-trivial, the authors provided quantitative evidence and a reasonable discussion. Given the methodological novelty and empirical gains, I view efficiency as an essential yet sufficient condition for future work, rather than a blocker. Overall, the paper makes clear technical contributions and provides insightful analysis that is likely to influence follow-up research on efficient diffusion-based tracking. I recommend acceptance.

**Limitations:**

Yes

**Paper Formatting Concerns:**

No concerns

**Quality:**

3

**Strengths And Weaknesses:**

### Strengths:

1.	The proposed approach is novel and conceptually interesting. It leverages video diffusion models in a previously unexplored way for tracking.

2.	The method achieves state-of-the-art performance without supervised training, demonstrating strong generalization capabilities.

3.	Extensive studies on different video and image diffusion models show the generalizability of the proposed method.

### Weaknesses:
1.	According to the results shown in Figure 6.a, the appearance feature (R_f) plays a negative role in motion tracking, but combining it with the motion feature (R_m) can bring some benefit. Please provide more insights about why this happens.
2.	High computational cost. Despite the superior performance, the high computational cost of the diffusion model may limit the real-time tracking.

---

> ### Author Rebuttal · Authors · 2025-07-31
>
> Thank you for your positive review and insightful feedback. We are glad that you find **our approach novel and conceptually interesting, which also achieves state-of-the-art tracking performance with strong generalization capabilities**. We truly appreciate your time and effort in reviewing our work, and we address your concerns below. Please let us know if any further information is needed.
>
> ### 1. Discussions on the appearance and motion features
>
> Thank you for your comment. When tracking similar-looking objects, motion features are most useful for determining object identity, while appearance features provide fine-grained details for precise segmentation. Once identity is determined, the remaining task is to leverage those appearance cues to refine the object mask. Therefore, appearance features also bring some benefit after combining with motion features, when tracking similar-looking objects. We will include these discussions in the revised paper.
>
>
> ### 2. Discussions on computation cost
>
> Thank you for your comment. Our method has 20% more computation time than generative model method DIFT on DAVIS videos and 89% than the popular discriminative model method SFC. The significant improvement in tracking accuracy is worth this additional computational cost.  We also discuss optimizing computational efficiency in Appendix B.4 and Appendix C.
>
>
> ### 3. Clarifying the visual elements in Figure 1
>
> Thank you for your suggestions. In Figure 1(c), the green and red masks represent segmentation maps of different objects. The boxes are used to draw the reader’s attention to specific regions. The blue, green, and red boxes highlight the ground truth regions, correctly predicted regions, and incorrectly predicted regions, respectively. We will follow your suggestions to revise the caption of Figure 1 to be more clear.

---

> > ### Comment · Reviewer_aXR3 · 2025-08-04
> >
> > Thanks very much for your detailed clarifications. I'll maintain my score.

---

### Official Review · Reviewer_ueSt · 2025-06-27

**Clarity:** 2
**Significance:** 2
**Originality:** 2
**Rating:** 4
**Confidence:** 4

**Summary:**

This paper tackles the problem of video label propagation, which aims to propagate the given pixel labels in the first video frame to the subsequent frames. The authors explore using pre-trained video diffusion model and image diffusion model to extract motion and appearance features, where no additional training is needed. The experimental results show good performance.

**Questions:**

1. The features extracted from diffusion models are interpolated to match input image size. Without further training, can the proposed method handle objects with irregular or complex shapes?

Please see weaknesses 1-4. My major concern is with the experiment settings.

**Ethical Concerns:**

["NO or VERY MINOR ethics concerns only"]

**Final Justification:**

After reading through the authors' response, most of my concerns and questions are answered. I'm inclined to accept this paper.

**Limitations:**

yes

**Quality:**

3

**Strengths And Weaknesses:**

Strengths:
1. The paper is easy to follow.
2. The provided analysis is thorough and insightful.
3. The proposed method shows good results on the compared benchmarks.

Weaknesses:
1. The proposed method shows good performance. Yet, many existing works have used pre-trained diffusion models for tracking and segmentation, e.g., [a,b,c]. Thus, it's not surprising that leveraging powerful video and image diffusion models can improve the results. However, as video label propagation is closely related to segmentation and tracking, it would be better to discuss and compare with similar works.
2. The compared methods in Tab. 1 seem outdated; the authors did not compare with methods after 2023.
3. The proposed pipeline involves multiple forward passes of video diffusion model and image diffusion model to perform the task. This would cause heavy overhead on computation and inference time, in my experience. Yet, there is no analysis on computation costs.
4. The authors emphasized "similar-looking objects" in the task and introduced two datasets (Youtube-Similar and Kubric-Similar), where the proposed method shows the most significant improvement over the compared methods. How are the videos in the introduced datasets selected? What's the proposed method's performance on other standard benchmarks (e.g., JHMDB)?



[a] Track4Gen: Teaching Video Diffusion Models to Track Points Improves Video Generation. CVPR 2025.
[b] Diff-Tracker: Text-to-Image Diffusion Models are Unsupervised Trackers. ECCV 2024.
[c] VidSeg: Training-free Video Semantic Segmentation based on Diffusion Models. CVPR 2025.

---

> ### Author Rebuttal · Authors · 2025-07-31
>
> Thank you for the valuable feedback. We are glad that the reviewer finds **our analysis thorough and insightful, while the proposed method achieves good results**. We truly appreciate your time and effort in reviewing our work, and we address your concerns below. Please let us know if any further information is needed.
>
> ### 1. Discussions on related work
>
> Thank you for suggesting these papers[a,b,c]. We will cite and discuss them in the revised paper.  We highlight that **our work significantly differs from these studies[a,b,c] as follows**.
>
>
> * Track4Gen [a] tackles point tracking by **training video diffusion models on labeled point trajectories**, whereas our work explores pretrained video diffusion models for object segmentation **without any tracking‑specific training**.
>
> * Diff‑Tracker [b] is built on **image diffusion models** with additional motion encoders to learn temporal cues. By contrast, our work directly explores the **built‑in motion of pretrained video diffusion models** without extra modules.
>
> * VidSeg [c] performs **instance‑agnostic** video semantic segmentation that cannot distinguish different objects in the same category. It also requires maintaining and updating an additional KNN classifier to learn temporal changes during tracking. By contrast, our approach **can distinguish even similar‑looking objects** without extra components.
>
> Apart from the above differences from [a,b,c], we highlight **our contributions go beyond performance improvements, with unique insights and findings which are not discussed in [a,b,c].**
>
> * Our work is the first to reveal that features from video diffusion models at early denoising stages can capture temporal motions.
> * Our work is the first to reveal that video diffusion models excel at tracking similar-looking objects without supervision.
>
> Notably, the studies [a,c] are published in June 2025 (CVPR 2025), after the NeurIPS 2025 submission due in May.  According to NeurIPS 2025 policy quoted as follows, *"For the purpose of the reviewing process, papers that appeared online after March 1st, 2025 will generally be considered "contemporaneous" in the sense that the submission will not be rejected on the basis of the comparison to contemporaneous work."*  these studies[a,c] are contemporaneous work to our submission.
>
> [a] Track4Gen: Teaching Video Diffusion Models to Track Points Improves Video Generation. CVPR 2025. [b] Diff-Tracker: Text-to-Image Diffusion Models are Unsupervised Trackers. ECCV 2024. [c] VidSeg: Training-free Video Semantic Segmentation based on Diffusion Models. CVPR 2025.
>
>
> ### 2. Comparison with recent self‑supervised methods
>
> Thank you for your comment. To the best of our knowledge, we have compared with the latest self-supervised baselines available at the time of submission. If the reviewer is aware of a relevant self-supervised method published recently, please let us know, and we would like to compare it.
>
> ### 3. Clarification on computation cost
>
> Thank you for your comment. We report the computation cost of our method in Section 5.3 (lines 290–293) and provide more details in Appendix B.4. Our method has 20% more computation time than generative model method DIFT on DAVIS videos and 89% than the popular discriminative model method SFC. We also discuss optimizing the computational efficiency of our method in Appendix B.4 and Appendix C.
>
> ### 4. Implementation details and evaluation results on three more standard datasets and tasks
>
> Thank you for your comment. We describe the selection process for the YouTube-Similar and Kubric-Similar datasets in Section 5.1 (lines 215–220). YouTube-Similar is manually created from the YouTube-VOS dataset by selecting videos that contain multiple objects with similar appearances. Kubric-Similar is a new dataset generated using the Kubric simulator, with randomized object colors, sizes, and motions. We include all generated videos as Kubric-Similar.  We will revise the paper for clarity.
>
>
>
> Following your suggestions, we conducted experiments for human pose tracking on the standard JHMDB dataset. We also evaluated on two additional standard datasets for video object segmentation: YouTube-VOS [1] and MOSE [2]. Experimental results in the following table demonstrate that **our method consistently improves tracking accuracy across diverse datasets and tracking tasks.** We will include these results in the revised paper.
>
>
> |          |  JHMDB  |    |
> |---------------|:---------:|:---------:|
> |               | PCK@0.1 ($\uparrow$) | PCK@0.2 ($\uparrow$) |
> | SFC           | 61.9    | 83.0      |
> | Spa-then-temp | 66.4    | 84.4    |
> | DIFT          | 63.4    | 84.3    |
> | **Ours**          | **68.3**    | **85.8**    |
>
>
>
> |  |      | Youtube-VOS     |  |      |  MOSE    | |
> |-------------|:------:|:------:|:------:|:------:|:------:|:------:|
> |             | J&Fm ($\uparrow$) | Jm ($\uparrow$)   | Fm ($\uparrow$)   | J&Fm ($\uparrow$) | Jm ($\uparrow$)   | Fm ($\uparrow$)   |
> | DIFT        | 70.5 | 68.2 | 72.7 | 34.5 | 28.9 | 40.1 |
> | **Ours**        | **71.1** | **68.9** | **73.4** | **35.6** | **30.1** | **41.1** |
>
>
> [1] Ning Xu, Linjie Yang, et. al. Youtube-vos: A large-scale video object segmentation benchmark. arXiv preprint arXiv:1809.03327, 2018.
>
> [2] Henghui Ding, Chang Liu, et.al. MOSE: A New Dataset for Video Object Segmentation in Complex Scenes. ICCV2023.
>
> ### 5. Discussions on the interpolation of segmentation maps
>
> Thank you for your comment. Following prior work, we first generate segmentation maps at the feature resolution and then interpolate these segmentation maps to the original image size. While this lower-resolution process may lose details for objects with irregular or complex shapes, it is a common challenge in existing methods. Our experiments on multiple datasets (e.g., Figure 5) empirically show that our approach is effective for tracking objects in real-world videos.

---

> > ### Comment · Reviewer_ueSt · 2025-08-02
> > **New comment**
> >
> > Thank you for prividing this rebuttal. I've read through the responses, which address most of my concerns.
> >
> > I acknowledge that this work is different from the mentioned works in the way they leverage diffusion models. But given that leveraging diffusion models for solving tasks, especially for finding "correspondence" or “matching” between objects, my point is that it may be better to discuss how the proposed mechanism is particularly designed for solving this particular task, e.g., what makes it better than other diffusion methods in solving this task.
> >
> > Regarding the datasets in the experiment, I agree with reviewer TBTL that Kubric-similar dataset seems to be a very toy dataset. The authors also didn't disclose any details about the "manual" selection of the videos in YouTube-Similar. This is still a concern to me as being able to distinguish "similar-looking objects" is highlighted as a major advantange in this paper. I'm also looking forward to the authors response toward questions raised by reviewer TBTL.

---

> ### Author Response · Authors · 2025-08-04
> **Thank you for your helpful suggestions for strengthening the paper!**
>
> Thank you for your valuable suggestions for strengthening this paper! We answer your questions below.
>
> ### 1. How does the proposed method make particular designs for the matching task?
>
> Thank you for your question. **The key contribution of our work is demonstrating that simply using representations from video diffusion models already enables the tracking of similar-looking objects, significantly outperforming all existing self-supervised methods.** Our method does not rely on tracking-specific training or complex designs.
>
> **As noted by Reviewer aXR3, "The proposed approach is novel and conceptually interesting. It leverages video diffusion models in a previously unexplored way for tracking."** While editing supervised training data or introducing tracking-specific modules can improve tracking performance, our work highlights that video diffusion models have already learned an inherent ability for tracking, and is worth publishing.
>
> ### 2. Contributions of Kubric-Similar dataset and Youtube-Similar dataset
>
> To study the principle that video diffusion models excel at tracking similar-looking objects, it is necessary to conduct a study on a controlled dataset. We build Kubric-Similar, **not to introduce complexity, but to disentangle the factors that are important for tracking**. Kubric-Similar allows us to **understand the weaknesses of existing self-supervised methods**, which over-rely on appearance features. We then find that once video diffusion models are pretrained, **their features can track identical-looking objects without tracking-specific supervision**. Kubric-Similar also enables **further analysis, such as our finding that video diffusion models capture motion cues in the early stages of the denoising process.**
>
> We also evaluate the problem on real-world videos. Since **there is no existing video object segmentation dataset focused on similar-object tracking**, we annotate videos from the YouTube-VOS dataset. We find that **all existing self-supervised methods often fail in these scenarios, while our method significantly narrows the gap between self-supervised and supervised approaches.**
>
> ### 3. Implementation details on Youtube-Similar
>
> Thank you for pointing this out. We clarify the selection process and creation rubric of the YouTube-Similar dataset as follows. To ensure quality, three graduate students were hired to manually select videos according to this rubric. Only the videos that all annotators find qualified are included in the final YouTube-Similar dataset.
>
> * **Object similarity.** To create a dataset with similar-looking objects, we first select videos that contain at least two objects belonging to the same category from YouTube-VOS.
>
> * **Video filtering.** We exclude static videos from the pool obtained in the first stage. In such videos, spatial appearance features can serve as a shortcut for tracking, which may influence our evaluation of the motion features learned in model representations.
>
> We will include these details in the revised paper.

---

> > ### Comment · Reviewer_ueSt · 2025-08-04
> > **Thank you for your response**
> >
> > Thank you for your response and the provided implementation details. Most of my concerns have been answered, and I will raise my rating to 4.

---

> > > ### Author Response · Authors · 2025-08-04
> > > **Thank you!**
> > >
> > > Thank you very much for reading our response and for deciding to raise your score. We truly appreciate your thoughtful feedback and valuable suggestions. We will carefully incorporate them into the revised version of the paper.
> > >
> > > Thank you again for your time and support!

---

### Official Review · Reviewer_TBTL · 2025-07-02

**Clarity:** 4
**Significance:** 4
**Originality:** 4
**Rating:** 5
**Confidence:** 3

**Summary:**

The paper studies the task of label propagation specifically for the challenging case of objects with similar appearance. They identified that the 3D UNet denoising models learned for generating videos with video diffusion models (pretext task) generate motion representations which can generalize to the task of label propagation (downstream task), highlighting the tracking capabilities of such models.

They generate frame features for each frame of a video by concatenating frame appearance features and frame motion features. For motion features, they add noise to a video by following the noise schedule of diffusion models. This noisy video is fed jointly (using video clips of fixed length) to a 3D-UNet model from which they extract frame features at the nth level.  For appearance features, they feed each noisy frame separately to a 2D UNet of an image diffusion model.

They propagate labels from the first frame to the current one by weighting the previous frame's pixel labels based on pixel similarity scores within the local neighborhood of the current frame's pixels.

For this study, they leverage pretrained diffusion models trained offline, and don’t pretrain them.

**Questions:**

- Can this work be extended to other video label propagation datasets? Traditionally, the video label propagation models are also evaluated on two other datasets (JHMDB[A] and VIP[B]) where the propagated labels are not only segmentation masks but human body parts or pose keypoints. Could the proposed method be extended to those scenarios?
- Could the author provide videos the following two videos (can be merged in one):
  - the segmentation label propagation to emphasize the quality of the tracker and
  - video extending figure 4, which would emphasize the temporal motion consistency of the feature extracted?
- Could the authors compare the results of Sam2 and Cutie models on their proposed datasets?

**Ethical Concerns:**

["NO or VERY MINOR ethics concerns only"]

**Final Justification:**

This paper claims that pretrained video diffusion models provide good representations that capture motion dynamics. They can be **used as is** to enrich representation only learned from image diffusion models. To highlight the benefits of their new proposed representation, they designed a specific use case where motion dynamics is important to learn: label propagation of similar appearance objects.

They introduce two new datasets:
* Kubric similar: a toy dataset where balls with similar appearance are moving to disambiguate the contribution of appearance and motion cues and **quantify if models can capture motion dynamics**. On this dataset, supervised models achieve almost perfection and the authors show that their proposed method is the first self-supervised method to be able to bridge the gap with SoTA supervised models (+ 80% compared with the best self supervised approach). They even surpass a less recent supervised method emphasizing that it relies more on appearance cues than on motion cues.
* YouTube-Similar: a real world dataset extracted from the YouTube VOS dataset. On this dataset they provide a +20% boost compared with the best performing self supervised method (and with SAM2 as the upper limit). If the gains are more limited than on the Kubric similar dataset, they are still significant in more complex scenarios where other factors (lightning, deformation of the object) that still rely on the appearance cue are also non-negligible. **By concatenating the appearance and motion feature maps they propose a simple integration of appearance and motion cues and emphasize the importance of both in the design of a good video representation**. Simple concatenation is not enough, but it is a valid starting point for this study.

Eventually, they also prove the validity of the proposed representation learning scheme on standard datasets with not only similar appearance objects for three label propagation tasks:
- VOS task with Davis, YTVOS and MOSE dataset
- Human body part propagation with the VIP dataset
- Human pose with the J-HMDB dataset

For all of those cases, the proposed method provides a boost in performance with regard to other self supervised models. However, it doesn’t bridge the gap with supervised models.

Regarding the time consumption of the proposed model, it is more consuming than other self supervised approaches (+20% to the most costly method) which can be balanced with the accuracy boost it provides. Nonetheless, given the exploratory component of this study, I would not take into account a lack of accuracy and time performance for accepting this paper (but this is a personal opinion, and I can completely understand diverging opinions).


Regarding my rating:
- The 5 rating is a pretty strong rating (looking at the text near the rating), and I cannot say that the evaluation conducted is excellent with strong confidence, because probably more could have been done before the rebuttal and still after the rebuttal (specifically on the VIP dataset). However, I believe this is a technically solid paper, and that representation learning of motion dynamics is an important research question, which justifies (in my opinion) the impact on at least one sub-area of AI. As a reader, I was very interested in this paper, and would be happy to discuss it with co-workers or the authors. I believe it deserves to be shared with the community.

**Limitations:**

yes

**Quality:**

3

**Strengths And Weaknesses:**

## Paper Strengths:

[An important study for the community]
- The task and the problem is well identified and motivated with a simple toy example. Tracking similar appearance objects is still a challenging task that is worth studying and which is understudied in supervised tasks such as VOS because of lacking datasets highlighting the challenge.

[New datasets contribution]
- Two datasets are going to be released upon acceptance of the paper (written in the Neurips paper checklist). Highlighting the challenges of tracking similar looking objects is an essential part of this paper, and releasing the two new datasets is a clear contribution that would benefit not only to the self-supervised community but would also benefit to the supervised community.

[Paper Clarity]
- Well written paper and well organised paper structure, with a clear motivation. Qualitative visualizations support the analysis, which strengthens the message.

[Qualitative analysis demonstrate the quality of the tracker]
- Figure 4 presents examples with not only similar appearance objects (e-f) but also challenging motion examples including motion blur (d), similar appearance background/foreground (b), body part occlusions (a), zoomed out to zoomed in objects (c ). Such results are impressive, and it would be interesting to confirm them with video results.

## Weaknesses:

[Clarification of the claim about video diffusion models]
- The study focuses on video diffusion models. What is the definition of a video diffusion model? is it to have a 3D Unet noise model? Because technically, a video diffusion model could be built on top of a 2D UNet noise model, or even have different architecture. The title makes a strong claim, whereas the experimental study (I2VGen-XL and Stable Video Diffusion) is only performed for 2 pretrained diffusion models using 3D-Unet.
- Specifically, l123, they introduce 2D-UNet, and l139-140, they mention temporal attention and 3D convolution to be essential for temporal information. A clarification paragraph at the end of section 4.1 is needed.

[Importance of the layer from which the representation is extracted]
- In the pseudo code in the appendix section, two parameters are mentioned: “Block index for features: nv (video diffusion) and ni (image diffusion)”, but I cannot find reference about them in the main paper. Which layers were selected in your model? Should it be the same layer for the motion and image features?

[Missing evaluations]
- It would be valuable to report performance on additional segmentation label propagation datasets (such as YouTube VOS[51], MOSE[C]) which contain but are not limited to similar appearance objects. This would better demonstrate the proposed model’s general effectiveness on the label propagation task for segmentation masks. Relying solely on the DAVIS-17 validation split, which is a small dataset with limited similar challenging examples, is insufficient for a comprehensive evaluation.
- Can this work be extended to other video label propagation datasets? Traditionally, the video label propagation models are also evaluated on two other datasets (JHMDB[A] and VIP[B]) where the propagated labels are not only segmentation masks but human body parts or pose keypoints. Could the proposed method be extended to those scenarios?

[Extension to the supervised trackers of VOS]
- The authors note that supervised trackers require extensive labeled data, yet the paper lacks evaluation of SoTA supervised models (SAM2 [35], Cutie [8], and others [9] [10]). Given that these models also struggle to distinguish between objects that have similar appearances, the proposed datasets would be valuable to quantify the performance of both self-supervised and supervised models on similar appearance object tracking. However, the absence of results for SAM2 and Cutie is a missed opportunity. Comparing supervised and self-supervised models—even if the latter underperform—would strengthen the study, as it addresses different, yet related, tasks.

[Video qualitative analysis]
- Missing video examples for a video analysis. Such an analysis would demonstrate the quality of the tracker.


## Minor weaknesses:
In the main method section, the appearance features are not described, but are described partially in the implementation details l191-193. It would benefit from being in the core method.


## Additional references:
[A] H. Jhuang, J. Gall, S. Zuffi, C. Schmid, and M. J. Black. Towards understanding action recognition.

[B] Qixian Zhou, Xiaodan Liang, Ke Gong, and Liang Lin. Adaptive temporal encoding network for video instance-level human parsing.

[C]  Henghui Ding,  Chang Liu,  Shuting He,  Xudong Jiang,  Philip H.S. Torr,  Song Bai MOSE: A New Dataset for Video Object Segmentation in Complex Scenes

---

> ### Author Rebuttal · Authors · 2025-07-31
>
> Thank you for your positive review and insightful suggestions. We are glad that **the reviewer recognizes our work as an important study for the community,  for highlighting the challenging and understudied task of tracking similar-looking objects**. We also appreciate the **positive comments on the contribution of new datasets, the well written paper with clear motivation,  and the impressive qualitative visualizations that demonstrate significant improvements across diverse video scenarios.** We truly appreciate your time and effort in reviewing our work, and we address your concerns below. Please let us know if any further information is needed.
>
> ### 1. Clarification on video diffusion models
>
> #### 1.1 Discussions on video diffusion models
>
> Thank you for your comment. Video diffusion models are often obtained by adding a temporal dimension to image diffusion models, using 3D architectures to capture temporal context. In this work, we study the widely used 3D UNet, which is built by inserting temporal layers into a 2D UNet. Our tracking method achieves significant improvement in tracking similar-looking objects based on these models. However, our tracking method is model‑agnostic: it relies on intermediate features from any pretrained video diffusion model, which may not be a 3D UNet. We will include these discussions in the revised paper.
>
>
> #### 1.2 Clarification on temporal layers in video diffusion models
>
> Thank you for pointing this out. The key to effectively tracking similar-looking objects is learning temporal motions among frames. We adopt the widely-used 3D UNet from video diffusion models, which is typically created by inserting the temporal layers into a 2D UNet. Temporal attention and 3D convolution are two successful and widely used temporal layers in video diffusion models. We will follow your suggestions to clarify this part in Section 4.1 of the revised paper.
>
>
> ### 2. Discussions on model layers for feature extraction
>
> Thank you for pointing this out. In Section 4.2 (lines 136-137), we define $n$ as the layer index for feature extraction. Following your suggestions, we will revise the paper to explicitly define $n_v$ and $n_i$ as the block indices for video and image diffusion models, respectively.
>
> Our framework is agnostic to specific layers.  Motion and appearance features can be taken from different layers, and the optimal layers for the two backbones do not need to be the same. We empirically extract motion features from the 3rd block of I2VGen‑XL and appearance features from the 8th block of ADM. These choices are noted in Section 4.5 (lines 189 and 194). For your reference, we report the effect of layers from video diffusion models on DAVIS dataset in the following table, which shows that features from 3rd block achieve the best tracking results.  We will include these experiments and discussions in the revised paper.
>
> | Block | J&Fm ($\uparrow$) | Jm ($\uparrow$)   | Fm ($\uparrow$)   |
> |-------|:------:|:------:|:------:|
> | 1     | 24.8 | 28.2 | 21.4 |
> | 2     | 47.6 | 52.7 | 42.5 |
> | 3     | 66.3 | 63.4 | 69.1 |
> | 4     | 31.5 | 27.2 | 35.8 |
>
>
> ### 3. Additional evaluations
>
> #### 3.1 Evaluation on additional video object segmentation datasets
>
> Thank you for your suggestions. We conducted experiments on Youtube VOS and MOSE datasets as you suggested. Experimental results in the following table show that our method consistently outperforms the state-of-the-art DIFT method in both datasets. **These results demonstrate the general effectiveness of our proposed methods across various video scenarios.** We will include these results in the revised paper.
>
>
> |  |      | Youtube-VOS     |  |      |  MOSE    | |
> |-------------|:------:|:------:|:------:|:------:|:------:|:------:|
> |             | J&Fm($\uparrow$) | Jm($\uparrow$)   | Fm($\uparrow$)   | J&Fm($\uparrow$) | Jm ($\uparrow$)  | Fm ($\uparrow$)  |
> | DIFT        | 70.5 | 68.2 | 72.7 | 34.5 | 28.9 | 40.1 |
> | **Ours**        | **71.1** | **68.9** | **73.4** | **35.6** | **30.1** | **41.1** |
>
>
> #### 3.2 Evaluation on additional tracking tasks
>
> Thank you for your comment. Yes, our method can be extended to other video label propagation tasks. Specifically, we conducted experiments on the human pose keypoints tracking task on JHMDB dataset, and report the experimental results in the following table. **Experimental results show that our approach significantly outperforms baseline  methods, demonstrating the effectiveness on tasks beyond video object segmentation.** We will include these results in the revised paper.
>
> |          | JHMDB   |    |
> |---------------|:-------------:|:---------:|
> |               | PCK@0.1($\uparrow$) | PCK@0.2($\uparrow$) |
> | SFC           | 61.9    | 83.0      |
> | Spa-then-temp | 66.4    | 84.4    |
> | DIFT          | 63.4    | 84.3    |
> | **Ours**          | **68.3**    | **85.8**    |
>
>
> ### 4. Comparison with supervised methods
>
> Thank you for your suggestions. We conducted experiments to evaluate state-of-the-art supervised models, including SAM2 [35], Cutiie [8],  XMem[9],  and STCN[10]. As the reviewer anticipated, self-supervised methods underperform the supervised ones. We will include these results in the revised paper.
>
>
> |            |               |       | Davis |       |       |  Youtube-Similar     ||      | Kubric-Similar| |
> |-----------------|---------------|:-------:|:-----------------:|:-------:|:-------:|:-------:|:----------------:|:------:|:------:|:------:|
> |                 |               | J&Fm ($\uparrow$)  | Jm  ($\uparrow$)              | Fm ($\uparrow$)    | J&Fm ($\uparrow$)  | Jm ($\uparrow$)    | Fm ($\uparrow$)             | J&Fm ($\uparrow$) | Jm ($\uparrow$)   | Fm ($\uparrow$)   |
> | Self-supervised | SFC           | 71.2  | 68.3            | 74.0    | 55.5  | 55.3  | 55.7           | 47.7 | 43.1 | 52.3 |
> |                 | Spa-Then-Temp | 74.1  | 71.1            | 77.1  | 59.6  | 59.2  | 60.1           | 48.9 | 44.0   | 53.8 |
> |                 | DIFT          | 75.7  | 72.7            | 78.6  | 60.7  | 59.8  | 61.7           | 55.1 | 52.7 | 57.6 |
> |                 | Ours          | 77.6  | 74.4            | 80.8  | 66.0    | 65.1  | 67.0             | 90.2 | 87.2 | 93.1 |
> | Supervised      | STCN           | 85.3  | 82.0              | 88.6  | 63.9  | 62.6  | 65.2           | 60.5 | 56.8 | 64.1 |
> |                 | XMem          | 86.2  | 82.9            | 89.5  | 82.1  | 79.9  | 84.4           | 98.8 | 97.9 | 99.7 |
> |                 | Cutie          | 87.2  | 84.3            | 90.1  | 87.3  | 84.7  | 89.8           | 99.1 | 98.2 | 100    |
> |                 | SAM2     | 89.8 | 86.7           | 92.9 | 86.9 | 84.1 | 89.6          | 99.4 | 98.8 | 100    |
>
>
> ### 5. Visualization of video examples
>
> Thank you for your suggestions. We have prepared the video examples you suggested, showing that our approach captures consistent motion features and consistently outperforms baseline methods across different frames.  However, due to this year's NeurIPS policy that prohibits external links or appendix materials during the rebuttal stage, we are afraid that we are unable to present video examples at this time. We will include them in the revised version of the paper to better demonstrate the effectiveness of our approach.
>
>
> ### 6. Implementation details about appearance features
>
> Thank you for your suggestions. We will revise the paper to describe the appearance features in the main method section. Specifically, we will incorporate the explanation currently in lines 191–193 into the method section (Section 4.3) to be more clear.

---

> > ### Comment · Reviewer_TBTL · 2025-08-01
> > **Thank you for providing a rebuttal answering my questions**
> >
> > Thank you for this rebuttal, I have read carefully the answers to my points but also to other reviewer comments. The authors have answered most of my points (and I acknowledge that the videos cannot be provided).
> >
> > ### Label Propagation
> > Regarding the task of label propagation, thank you for providing numbers for JHMBD (which was also pointed out by other reviewers). Is it possible to adapt the method for VIP as well? What prevents it otherwise (except for limited time constraints which is a valid point).
> >
> > ### Computational cost:
> > This is a valid point raised by other reviews, and the authors have answered it quantitatively. To mitigate the higher computational cost, I would argue that the point of the paper is not to provide a new tracker that would surpass all the benchmarks (including accuracy and the computational cost) but it is to provide a study of the potential of pretrained diffusion models for representation learning and specifically concerning label propagation of similar-appearance objects.
> >
> > ### Comparison with supervised-tracking methods:
> > I thank the authors for providing results for different VOS models on the two newly introduced datasets.
> >
> > It appears that the Kubric-similar dataset is a very toy dataset, for which the recent VOS models almost achieve perfection. What is striking though is that the proposed self-supervised model almost bridges the gap between self supervised and supervised models. Could the authors provide an analysis detailing the success-cases and failure cases? Is there any importance of the size of the object, the occlusions between objects? Is it performing well on the boundaries of the objects? What makes the results below XMem, Cutie and Sam2?
> >
> > Concerning real-world examples, it is an interesting dataset. First it highlights that the Cutie model slightly outperforms SAM2 without being trained on a very large annotated dataset, and using a well-designed object centric approach. Second, it also emphasizes that the proposed approach doesn’t bridge the gap but still makes significant improvements compared with other self-supervised approaches. Again, it would be very interesting to have an analysis of the success/failure cases, the importance of the size of the object, the occlusions, the boundaries (and other important factors) to have some insights of what is missing.
> >
> > ### Comparisons on YT-VOS and MOSE:
> > Thank you for providing the results and it appears that the proposed method consistently improves the results over both datasets, but are not bridging the gap with supervised models, which could be explained by the fact that there are more non similar appearance objects than similar appearance objects.
> > To go further, could the authors provide numbers for YT-VOS without the examples in YT-similar? It would highlight that the approach doesn’t harm the results when there are few (can the authors discuss about few or none?) similar appearance objects.
> > An extension of YT similar could be done on MOSE as well, but this is a task that goes way beyond the scope of the rebuttal. Still, it is a be disappointing that the methods doesn't perform better on MOSE. Can the authors provide some insights/analysis about why?

---

> > > ### Comment · Reviewer_TBTL · 2025-08-01
> > > **Clarification of possible confusion**
> > >
> > > Just to clarify my comments, when I ask for an analysis of success / failure cases, it means with respect to the supervised models. Where does the current model doesn't match the VOS models?

---

> ### Author Response · Authors · 2025-08-04
> **Thank you for your helpful suggestions for strengthening the paper! (Part 1/2)**
>
> Thank you for your valuable suggestions for strengthening this paper! We answer your questions below.
>
> ### 1. Evaluation on VIP dataset
>
> Thank you for your suggestions. Due to the limited time, we did not provide the results on VIP dataset during the first rebuttal period. Following your suggestions, we conducted experiments to evaluate our method on VIP. Experimental results in the following table show that our method outperforms baseline models on VIP dataset. We will include these results in the revised paper.
>
> |   Method       |  mIoU ($\uparrow$) |
> |---------------|------|
> | SFC           | 38.4 |
> | Spa-then-temp | 41.0 |
> | DIFT          | 43.7 |
> | **Ours**          | **44.2** |
>
>
>
> ### 2. Contributions of Kubric-Similar dataset and Youtube-Similar dataset
>
> To study the principle that video diffusion models excel at tracking similar-looking objects, it is necessary to conduct a study on a controlled dataset. We build Kubric-Similar, **not to introduce complexity, but to disentangle the factors that are important for tracking**. Kubric-Similar allows us to **understand the weaknesses of existing self-supervised methods**, which over-rely on appearance features. We then find that once video diffusion models are pretrained, **their features can track identical-looking objects without tracking-specific supervision**. Kubric-Similar also enables **further analysis, such as our finding that video diffusion models capture motion cues in the early stages of the denoising process.**
>
> We also evaluate the problem on real-world videos. Since **there is no existing video object segmentation dataset focused on similar-object tracking**, we annotate videos from the YouTube-VOS dataset. We find that **all existing self-supervised methods often fail in these scenarios, while our method significantly narrows the gap between self-supervised and supervised approaches.**
>
>
> ### 3. Comparison with supervised tracking methods
>
> Thank you for your suggestions. We compare the tracking results of **our self-supervised method** with supervised approaches below.
>
> **3.1 Kubric-Similar**
>
> **Our work significantly improves the tracking accuracy of state-of-the-art self-supervised methods on Kubric-Similar by 35.1%** (see Table 1 in the paper). Visualization results also show that **our method successfully distinguishes identical-looking objects with varying colors, sizes, and motions, while all prior self-supervised methods fail**. One example is visualized in Figure 2 of the paper.  We further analyze key factors and compare with supervised methods below.
>
> * **Object sizes.** Kubric-Similar includes moving balls of random sizes generated by the Kubric simulator. Despite its best performance among self-supervised approaches, visualizations show that our method can still be improved when compared to supervised methods. For example, when tracking small objects, we observe some pixels belonging to small objects may be misclassified as background.
>
> To evaluate the impact of object size, we sort videos in Kubric-Similar by object size and split them evenly into three subsets: Large, Medium, and Small. As shown in the table below, our method performs best on large objects and worse on smaller ones. Supervised methods also show a similar trend, highlighting the general challenge of small objects in our Kubric-Similar. Notably, the performance gap between our method and SAM2 increases from 10.6% (Large) to 14.3% (Small), suggesting that object size is an important factor influencing tracking accuracy.
>
>
> || Model           | Large         | Medium  | Small |
> |------------------|------------------|---------------|---------|-------|
> | Self-supervised  | Spa-then-temp | 43.5    | 44.1  | 44.3 |
> |                  | DIFT           | 58.4    | 50.5  | 49.2 |
> |                  | Ours          | 88.8    | 88.2  | 84.6 |
> | Supervised       | XMem          | 98.6    | 98.6  | 96.4 |
> |                  | Cutie         | 98.9    | 98    | 97.8 |
> |                  | SAM2          | 99.4    | 99.2  | 98.9 |
>
>
> * **Boundary predictions.** Another performance gap with supervised methods lies in inaccurate boundaries in some cases. For example, with spatially adjacent objects, a small number of pixels near the object boundary may be mislabeled as belonging to the wrong object.
>
> * **Occlusion.** We find that our method outperforms self-supervised approaches in handling occluded objects, achieving a $J_m$ of 86.2% compared to 59.5% by DIFT. However, it still underperforms the supervised method SAM2, which reaches 97.5%.

---

> ### Author Response · Authors · 2025-08-04
> **Thank you for your helpful suggestions for strengthening the paper! (Part 2/2)**
>
> **3.2 Youtube-Similar**
>
> **Our method significantly outperforms prior self-supervised methods in tracking similar-looking objects in real-world videos**, as shown in Figures 1 and 5 of the main paper. We further compare it with supervised approaches below.
>
> * **Object sizes.** While our method effectively distinguishes different objects, it underperforms supervised methods on small objects. This is partly because small objects occupy fewer pixels in the reduced-resolution feature space, making it harder to preserve their information.
>
> * **Boundary predictions.** When our method underperforms the supervised counterpart, one reason is that although we predict the correct identity, the object boundary is not clear.  We believe this is partly due to the resolution reduction in the feature used for propagation, which may cause the loss of fine-grained details near object edges.
>
> * **Occlusion.**  Our method outperforms self-supervised baselines in handling occlusion. As shown in Figure 1 of the main paper, where one deer is occluded by another during movement, our method correctly distinguishes object identities after occlusion, whereas prior self-supervised methods fail. We find that supervised methods outperform ours in some cases because they capture fine-grained details more effectively, such as the legs of the deer in Figure 1.
>
> We will include these discussions and visualizations in the revised paper.
>
> ### 4. Discussions on Youtube-VOS and MOSE
>
> Thank you for your suggestions. We report the results on YouTube-VOS videos that do not contain similar-looking objects in the following table. The results show that our method outperforms the state-of-the-art DIFT method in non-similar object scenarios.
>
> | Model | J&F_m($\uparrow$) | J_m($\uparrow$)  | F_m($\uparrow$)  |
> |-------|-------|------|------|
> | DIFT  | 70.9  | 68.5 | 73.2 |
> | **Ours**  | **71.4**  | **69.1** | **73.7** |
>
> Experimental results show that our method outperforms state-of-the-art self-supervised DIFT method on MOSE, as shown in the following table for your reference.
>
> | Model | J&F_m($\uparrow$) | J_m($\uparrow$)  | F_m($\uparrow$)  |
> |-------|-------|------|------|
> | DIFT        |  34.5 | 28.9 | 40.1 |
> | **Ours**        |  **35.6** | **30.1** | **41.1** |

---

> > ### Comment · Reviewer_TBTL · 2025-08-04
> >
> > Very nice, I believe that a revised version of the paper containing the answers provided to my points and the questions raised by reviewer ueSt will be stronger, and I am pleased with the answers. I will keep my rating as is.
> > Thank you for your hard work!

---

> > > ### Author Response · Authors · 2025-08-04
> > > **Thank you!**
> > >
> > > We are sincerely grateful for your thoughtful feedback and your continued support for the acceptance of this paper.  Following your suggestions, we will incorporate our responses to your points and the questions raised by reviewer ueSt into the revised version of the paper.
> > >
> > > Thank you again for your time and support!

---

### Official Review · Reviewer_HZ3G · 2025-07-03

**Clarity:** 3
**Significance:** 2
**Originality:** 3
**Rating:** 4
**Confidence:** 5

**Summary:**

This paper proposes a simple yet effective method for visual tracking, especially in the scenario of tracking similar-looking objects.The method employs the learned features by pre-trained video diffusion model in the early denoising stage as the motion feature based on the observation that such features excel at capturing motion dynamics.

The idea is simple yet interesting and the experiments show the effectiveness of the method.

**Questions:**

Refer to the weaknesses.

**Ethical Concerns:**

["NO or VERY MINOR ethics concerns only"]

**Final Justification:**

Thanks for the response which has addressed most of my concerns. Nevertheless, I am still not quite satisfied about the lack of a more convincing demonstration of the basic observation/assumption that the learned features by the pretrained video diffusion models at early denoising stage have the innate ability of capturing motion dynamics.
Based on the response, I am glad to raise my score to 4.

**Limitations:**

Yes.

**Quality:**

2

**Strengths And Weaknesses:**

Strengths:
1. The idea is simple yet interesting.
2. The paper is presented well.
3. The experiments demonstrate the effectiveness of the method.

Weaknesses:
1. The paper is based on the observation/assumption that the learned features by the pretrained video diffusion models at early denoising stage have the innate ability of capturing motion dynamics. Is there any theoretical evidence for this apart from the experimental observations?
2. I don't think it is a fair comparison to compare the proposed method with only self-supervised methods since the proposed method is built upon the pre-trained diffusion models for both videos and images. Is it possible to compare it with supervised trackers or develop a supervised version of the proposed method and then make a comparison with existing supervised trackers?
3. How about the performance of the proposed method on typical/routine tracking scenarios/datasets?
4. I am curious about the efficiency of the proposed method. Extracting features during inference requires the inference of both a pretrained video diffusion model and a pretrained image diffusion model, which seems computationally expensive.
5. I doubt the effectiveness of the weighting factor 'lambda' in Equation 5, since the R_m and R_a are extracted from two individual feature space.

---

> ### Author Rebuttal · Authors · 2025-07-31
>
> Thank you for the valuable feedback. We are glad that the reviewer finds **our proposed tracking method interesting and effective, supported by experiments**. We truly appreciate your time and effort in reviewing our work, and we address your concerns below. Please let us know if any further information is needed.
>
>
>   ### 1. Paper scope and contribution of our empirical findings
>
>  Thank you for your comment. As you pointed out, our work empirically demonstrates that features learned by pretrained video diffusion models at early denoising stages capture motion dynamics. We highlight that **one of our contributions is to be the first to report this property and validate its effectiveness for tracking through empirical experiments.** Theoretical derivation is beyond the scope of this work, and rigorous theory remains an open challenge for the deep learning field.
>
>
> ### 2. Our method is fully self-supervised
>
> Thank you for your comment. **Our work uses diffusion models trained without any tracking-specific labeled data, and is self-supervised in terms of tracking.**
>
> As you suggested, we ran experiments comparing our approach with state‑of‑the‑art supervised methods, including SAM2[1], Cutie[2], XMem[3], and STCN[4]. Self‑supervised methods often underperform their supervised counterparts. Nevertheless, **our method achieves the best tracking accuracy among self‑supervised approaches and significantly narrows the gap between self‑supervised and supervised methods**. We will include these results in the revised paper.
>
>
> |            |               |      | Davis|      |    | Youtube-Similar  ||
> |-----------------|---------------|:------:|:------:|:------:|:------:|:------:|:------:|
> |                 |               | J&Fm ($\uparrow$) | Jm  ($\uparrow$)            | Fm ($\uparrow$)  | J&Fm ($\uparrow$) | Jm ($\uparrow$)   | Fm ($\uparrow$)  |
> | Self-supervised | SFC           | 71.2 | 68.3            | 74.0   | 55.5 | 55.3 | 55.7 |
> |                 | Spa-Then-Temp | 74.1 | 71.1            | 77.1 | 59.6 | 59.2 | 60.1 |
> |                 | DIFT          | 75.7 | 72.7            | 78.6 | 60.7 | 59.8 | 61.7 |
> |                 | Ours          | 77.6 | 74.4            | 80.8 | 66.0   | 65.1 | 67.0   |
> | Supervised      | STCN          | 85.3 | 82.0              | 88.6 | 63.9 | 62.6 | 65.2 |
> |                 | XMem          | 86.2 | 82.9            | 89.5 | 82.1 | 79.9 | 84.4 |
> |                 | Cutie         | 87.2 | 84.3            | 90.1 | 87.3 | 84.7 | 89.8 |
> |                 | SAM2          | 89.8 | 86.7            | 92.9 | 86.9 | 84.1   | 89.6 |
>
>
>
> [1] Nikhila Ravi,  Valentin Gabeur, et al. Sam 2: Segment anything in images and videos. ICLR 2025.
>
> [2] Ho Kei Cheng, Seoung Wug Oh, Brian Price, Joon-Young Lee, and Alexander Schwing. Putting the object back into video object segmentation. CVPR 2024.
>
> [3] Ho Kei Cheng and Alexander G Schwing. Xmem: Long-term video object segmentation with an atkinson327 shiffrin memory model. ECCV 2022.
>
> [4] Ho Kei Cheng, Yu-Wing Tai, and Chi-Keung Tang. Rethinking space-time networks with improved memory coverage for efficient video object segmentation. NeurIPS 2021.
>
> ### 3. Consistent improvement across three additional standard datasets and tasks
>
> Thank you for your suggestions. In the main paper, we have already shown significant performance improvements on three datasets, including standard DAVIS, YouTube-Similar, and Kubric-Similar.
>
> Following your suggestions, we conducted experiments on three additional, widely-used typical datasets and tasks, including:
> * Standard datasets for video object segmentation:  YouTube-VOS [5] and MOSE [6].
> * Standard dataset for human pose tracking: JHMDB [7].
>
> Experimental results in the following table demonstrate that **our method consistently improves tracking accuracy across diverse datasets and tracking tasks.** We will include these results in the revised paper.
>
> |  |      | Youtube-VOS     |  |      |  MOSE    | |
> |-------------|:------:|:------:|:------:|:------:|:------:|:------:|
> |             | J&Fm ($\uparrow$) | Jm ($\uparrow$)   | Fm ($\uparrow$)   | J&Fm ($\uparrow$) | Jm ($\uparrow$)   | Fm ($\uparrow$)   |
> | DIFT        | 70.5 | 68.2 | 72.7 | 34.5 | 28.9 | 40.1 |
> | **Ours**        | **71.1** | **68.9** | **73.4** | **35.6** | **30.1** | **41.1** |
>
>
> |          |  JHMDB  |    |
> |---------------|:---------:|:---------:|
> |               | PCK@0.1 ($\uparrow$) | PCK@0.2 ($\uparrow$) |
> | SFC           | 61.9    | 83.0      |
> | Spa-then-temp | 66.4    | 84.4    |
> | DIFT          | 63.4    | 84.3    |
> | **Ours**          | **68.3**    | **85.8**    |
>
>
> [5] Ning Xu, Linjie Yang, et. al. Youtube-vos: A large-scale video object segmentation benchmark. arXiv preprint arXiv:1809.03327, 2018.
>
> [6] Henghui Ding, Chang Liu, et.al. MOSE: A New Dataset for Video Object Segmentation in Complex Scenes. ICCV2023.
>
> [7] Hueihan Jhuang, Juergen Gall, et.al. Towards understanding action recognition. ICCV2013.
>
> ### 4. Clarification on computation cost
>
> Thank you for your comment. We report the computation cost of our method in Section 5.3 (lines 290–293) and provide more details in Appendix B.4. Our method has 20% more computation time than generative model method DIFT on DAVIS videos and 89% than the popular discriminative model method SFC. We also discuss optimizing the computational efficiency of our method in Appendix B.4 and Appendix C.
>
>
> ### 5. Discussions on fusion weight factor
>
> Thank you for your comment. We conduct experiments to investigate the effect of the weighting factor $\lambda$ in Section 5.3 (lines 270–281). Experimental results show that, across a range of fusion weight $\lambda$, our fused representation consistently outperforms using either motion or appearance features alone in real‑world videos. These results demonstrate the effectiveness of integrating features from video and image diffusion models through a simple weighted fusion scheme.

---

> > ### Comment · Reviewer_HZ3G · 2025-08-06
> > **Final score**
> >
> > Thanks for the response which has addressed most of my concerns. Nevertheless, I am still not quite satisfied about the lack of a more convincing demonstration of the basic observation/assumption that the learned features by the pretrained video diffusion models at early denoising stage have the innate ability of capturing motion dynamics. Based on the response, I am glad to raise my score to 4.

---

> > > ### Author Response · Authors · 2025-08-06
> > > **Thank you!**
> > >
> > > We sincerely thank you for reading our response and for deciding to raise your score. We greatly appreciate your valuable suggestions and will carefully incorporate them into the revised version of our paper.
> > >
> > > Thank you again for your time and support!

---

### Decision · Program_Chairs · 2025-09-17

**Decision:**

Accept (poster)

**Comment:**

This paper addresses the problem of tracking objects with similar appearances. The authors leverage pre-trained video and image diffusion models to extract motion and appearance features without additional training. By combining these features, the method enables effective pixel-level label propagation across frames. Experiments demonstrate consistent improvements over existing approaches on multiple benchmarks.

All reviewers are generally positive about this paper. However, two minor concerns were raised after the rebuttal: The first concern is the lack of stronger theoretical justification for why features learned by pre-trained video diffusion models in the early denoising stage naturally capture motion dynamics. As clarified in the authors’ response, one of the main contributions of this work is to be the first to observe and empirically validate this property for tracking. While a theoretical derivation would further strengthen the paper, it is beyond the intended scope. The AC finds this acceptable. The second concern relates to the higher computational cost compared to other self-supervised approaches. However, as Reviewer TBTL noted, this overhead is offset by the clear accuracy improvements provided by the method.

Based on the overall positive feedback and the minor nature of the remaining concerns, the AC recommends acceptance.